



# Chemical composition, structures, and light absorption of N-containing
# aromatic compounds emitted from burning wood and charcoal in
# household cookstoves
Mingjie Xie[1], Zhenzhen Zhao[1], Amara L. Holder[2], Michael D. Hays[2], Xi Chen[2], Guofeng Shen[3],
James J. Jetter[2], Qin'geng Wang[4]
[1]Collaborative Innovation Center of Atmospheric Environment and Equipment Technology,
Jiangsu Key Laboratory of Atmospheric Environment Monitoring and Pollution Control, School
of Environmental Science and Engineering, Nanjing University of Information Science &
Technology, 219 Ningliu Road, Nanjing 210044, China
[2]Center for Environmental Measurement and Modeling, Office of Research and Development, U.S.
Environmental Protection Agency, 109 T.W. Alexander Drive, Research Triangle Park, NC 27711,
USA
[3]Laboratory for Earth Surface Processes, College of Urban and Environmental Sciences, Peking
University, Beijing 100871, China
[4]State Key Laboratory of Pollution Control and Resource Reuse, Nanjing University, Nanjing
210023, China
*Correspondence to:* Mingjie Xie (mingjie.xie@colorado.edu; mingjie.xie@nuist.edu.cn)
Tel: +1-18851903788;
Fax: +86-25-58731051;
Mailing address: 219 Ningliu Road, Nanjing, Jiangsu, 210044, China





**Abstract**

N-containing aromatic compounds (NACs) are an important group of light-absorbing
molecules in the atmosphere. They are often observed in combustion emissions, but their chemical
formulas and structural characteristics remain uncertain. In this study, red oak wood and charcoal
fuels were burned in cookstoves using the standard water boiling test (WBT) procedure.
Submicron aerosol particles in the cookstove emissions were collected using quartz ($Q_f$) and
polytetrafluoroethylene (PTFE) filter membranes positioned in parallel. A back-up quartz filter
($Q_b$) was also installed downstream of the PTFE filter to evaluate the effect of sampling artifact on
NACs measurements. Liquid chromatography-mass spectroscopy (LC-MS) techniques identified
seventeen NAC chemical formulas in the cookstove emissions. The average concentrations of total
NACs in $Q_b$ samples ($0.37 \pm 0.31 - 1.78 \pm 0.78$ µg m$^{-3}$) were greater than 50% of those observed
in the $Q_f$ samples ($0.47 \pm 0.40 - 3.54 \pm 1.63$ µg m$^{-3}$), and the $Q_b$ to $Q_f$ mass ratios of individual
NACs had a range of $0.02 - 2.71$, indicating that the identified NACs might have substantial
fractions remaining in the gas-phase. In comparison to other sources, cookstove emissions from
red oak or charcoal fuels did not exhibit unique NAC structural features, but had distinct NACs
composition. However, before identifying NACs sources by combining their structural and
compositional information, the gas-particle partitioning behaviors of NACs should be further
investigated. The average contributions of total NACs to the light absorption of organic matter at
$\lambda = 365$ nm ($1.10 - 2.58\%$) in $Q_f$ samples are much lower than those in $Q_b$ samples ($10.7 - 21.0\%$).
These results suggest that more research is needed to understand the chemical and optical
properties of gaseous chromophores and heavier molecular weight (e.g., MW > 500 Da) entities
in particulate matter.




## 1 Introduction


In the developing world, 2.8 billion people burn solid fuels in household cookstoves for
domestic activities such as heating and cooking (Bonjour et al., 2013). A variety of gaseous and
particulate phase pollutants ⸺ carbon monoxide (CO), nitrogen oxides ($NO_X$), volatile organic
compounds (VOCs), fine particulate matter with aerodynamic diameter $\leq$ 2.5 µm ($PM_{2.5}$), black
carbon (BC), organic carbon (OC), etc. ⸺ are emitted from cookstoves largely due to incomplete
combustion (Jetter et al., 2012; Shen et al., 2012; Wathore et al., 2017). In China, the relative
contributions of residential coal and biomass burning (BB) to annual $PM_{2.5}$ emissions decreased
from 47% (4.32 Tg) in 1990 to 34% (4.39 Tg) in 2005 due to the growth in industrial emissions
(Lei et al., 2011). Although, more than half of BC (> 50 %) and OC (> 60 %) emissions are
attributed to residential coal and BB in both China and India (Cao et al., 2006; Klimont et al., 2009;
Lei et al., 2011).
Household solid fuel combustion is a leading human health risk, especially for women and
children who tend to spend more time indoors than men (Anenberg et al., 2013). Estimates show
that exposures to $PM_{2.5}$ from domestic solid fuel combustion caused 3.9 million premature deaths
and ~4.8% of lost healthy life years (Smith et al., 2014). In addition, the emissions of carbonaceous
aerosols from cookstoves can affect the Earth's radiative balance by absorbing and scattering
incoming solar radiation (Lacey and Henze, 2015; Aunan et al., 2009). BC is the most efficient
light absorber in the atmosphere, while the total aerosol absorption, including that from OC, is still
highly uncertain (Yang et al.,2009; Park et al., 2010; Feng et al., 2013; Wang et al., 2014; Tuccella
et al., 2020). Multiple field and laboratory studies have demonstrated that OC in both primary PM
emissions (e.g., biomass and fossil fuel combustions) and secondary organic aerosol (SOA) feature
a range of absorptivity in the near ultraviolet (UV) and short visible wavelength regions



(Nakayama et al., 2010; Forrister et al., 2015; Lin et al., 2015; De Haan et al., 2017; Xie et al.,
2017a, b, 2018). The light absorbing OC fraction is often referred to as "brown carbon" (BrC).
Unlike open BB (e.g., forest, grassland, and cropland fires) —— one of the most important primary
sources for organic aerosols (Bond et al., 2004) —— the light absorption of BrC from household
cookstove emissions is rarely investigated. Sun et al. (2017) found that the BrC absorption from
residential coal burning accounted for 26.5% of the total aerosol absorption at 350~850 nm. BrC
from wood combustion in cookstoves has a greater mass specific absorption than that from open
BB over the wavelength range of 300 – 550 nm (Xie et al., 2018). These results suggest that
cookstove emissions may also be an important BrC source, which needs to be accounted for
separately from open BB.
Organic molecular markers (OMMs) are commonly used in receptor-based source
apportionment of carbonaceous aerosols (Jaeckels et al., 2007; Shrivastava et al., 2007; Xie et al.,
2012). Polycyclic aromatic hydrocarbons (PAHs) and their derivatives are a group of OMMs with
light absorption properties dependent on ring number or the degree of conjugation (Samburova et
al., 2016). As discussed in Xie et al. (2019), PAHs are generated from a multitude of combustion
processes (e.g., BB, fossil fuel combustion) (Chen et al., 2005; Riddle et al., 2007; Samburova et
al., 2016), and their ubiquitous nature makes them less than ideal OMMs for BrC source attribution.
Because of the specific toxicological concern raised by PAHs —— they are mutagenic and
carcinogenic [International Agency for Research on Cancer (IARC), 2010] —— source emission
factors, ambient levels, and potential health effects of PAHs are investigated exhaustively
(Ravindra et al., 2008; Kim et al., 2013). Similar to PAHs, N-containing aromatic compounds
(NACs) are a group of BrC chromophores commonly detected in ambient PM and source
emissions. Zhang et al. (2013) and Teich et al. (2017) calculated the absorption of individual NACs



in aqueous extracts of ambient PM, the total of which explained ~3% of the bulk extract absorption
at 365 – 370 nm. With the same approach, Xie et al. (2017a, 2019) found that the absorbance due
to NACs in BB or secondary OC was 3 – 10 times higher than their mass contributions. Lin et al.
(2016, 2017) estimated an absorbance contribution of 50 – 80% from NACs in BB OC directly
from their high-performance liquid chromatography (HPLC)/photodiode array (PDA) signals,
which are subject to considerable uncertainty due to the co-elution of other BrC chromophores
(e.g., PAHs and their derivatives). These results indicate that NACs are strong BrC chromophores,
but the estimation of their contributions to BrC absorption depends largely on how well they are
chemically characterized. Nitrophenols, methyl nitrophenols, nitrocatechols and methyl
nitrocatechols (including isomers) are typical atmospheric NACs (Claeys et al., 2012; Desyaterik
et al., 2013; Zhang et al., 2013). These NACs can be generated from BB (Lin et al., 2016, 2017;
Xie et al., 2019), fossil fuel combustion (Lu et al., 2019), and the reactions of aromatic volatile
organic compounds (VOCs) with reactive nitrogen species (e.g., $NO_X$) (Xie et al., 2017a), and are
not unique to specific sources (e.g., BB). By using a HPLC interfaced to a diode array detector
(DAD) and quadrupole (Q) time-of-flight mass spectrometer (ToF-MS), Xie et al. (2019) found
that BB NACs contain methoxy and cyanate groups. Nitronaphthol, nitrobenzenetriol, and methyl
nitrobenzenetriol are characteristic NACs for $NO_X$-based chamber reactions of naphthalene,
benzene, and *m*-cresol, respectively (Xie et al., 2017a). Yet, few studies have investigated the
composition of NACs from household cookstove emissions (Fleming et al., 2018; Lu et al., 2019).

The present study aims to characterize NACs in $PM_{2.5}$ from burning red oak and charcoal in

a variety of cookstoves and calculate their contributions to bulk OC absorption. The absorption of
OC in solvent extracts of cookstove emissions were measured in our previous work (Xie et al.,
2018). Presently, NACs are identified and quantified using an earlier described HPLC/DAD-Q-



ToF-MS system. In addition, the NACs adsorbed on a backup quartz filter downstream of a
polytetrafluoroethylene (PTFE) membrane filter are analyzed, to evaluate the potential for
sampling artifacts of NACs in $PM_{2.5}$. This work unveils BrC composition at a molecular level and
increases the understanding of BrC chromophores and their sources. It also shows that further
identification of large molecules (e.g., > 500 Da) may better explain BrC absorption in the particle
phase and that understanding the light absorption of gaseous chromophores is important for the
future.
**2 Methods**
*2.1. Cookstove emissions sampling*

Details of the cookstove emission test facility, fuel-cookstove combinations, water boiling

test (WBT) protocol, and $PM_{2.5}$ emissions sampling were described previously in Jetter et al. (2012)
and Xie et al. (2018). Briefly, the cookstove emissions tests were performed at the U.S. EPA
cookstove test facility in Research Triangle Park, NC. Three fuels (red oak wood, lump charcoal,
and 1-K kerosene) were burned in fuel-specific cookstoves. Due to the limited sample number (*n*
= 6) and low OC emissions from kerosene burning, only red oak and charcoal burning samples
were used for NACs analysis. Low moisture (~10%) oak and charcoal fuels were burned with five
specific-designed cookstove types (Tables S1 and S2); high moisture (~30%) oak fuels were
burned in one cookstove (Jiko Poa). Emissions tests for each fuel-cookstove combination were
performed in triplicate. The WBT protocol (version 4) (Global Alliance for Clean Cookstoves,
2014) is designed to measure cookstove power, energy efficiency and fuel use and utilizes cold-
start (CS) high power, hot-start (HS) high power, and simmer (SIM) low power phases. Gaseous
pollutant (e.g., CO, methane ($CH_4$)) emissions were monitored continuously, and $PM_{2.5}$ filter
samples were collected during each test phase of the WBT protocol. A quartz-fiber filter ($Q_f$) and



a PTFE membrane filter positioned in parallel collected $PM_{2.5}$ isokinetically at a flow rate of 16.7
L min$^{-1}$. Adsorption artifact was evaluated using a quartz-fiber back-up filter ($Q_b$) installed
downstream of the PTFE filter during $PM_{2.5}$ sampling.
*2.2. Chemical analysis*
The OC and elemental carbon (EC) emissions and UV-Vis light absorption properties (BrC)
of methanol-extracted cookstove particles were reported in Xie et al. (2018). Details of the method,
sample selection, and measurement results are summarized in supplementary information (Text
S1, Tables S1 and S2). The $Q_f$ and $Q_b$ sample extraction and subsequent analysis for NACs were
conducted as described in Xie et al. (2019). In brief, an aliquot of each filter sample was pre-spiked
with 250 ng nitrophenol-d4 (internal standard) and extracted ultrasonically twice for 15 min in 3-
5 mL of methanol. After filtration (30 mm diameter. ×0.2 μm pore size, PTFE filter), the extract
volume was reduced to ~500 μL with rotary evaporation prior to HPLC/DAD-MS (Q-ToF)
analysis. The NACs targeted in this work were chromatographed using an Agilent 1200 Series
HPLC equipped with a Zorbax Eclipse Plus C18 column (2.1 mm × 100 mm, 1.8 μm particle size;
Agilent Technologies). The gradient separation was performed using water (eluent A) and
methanol (eluent B) containing 0.2% acetic acid (v/v) with a total flow rate of 0.2 mL min$^{-1}$. The
eluent B fraction was held at 25% for 3 min, increased to 100% over the next 7 min, where it was
held for 22 min, and then returned to 25% over 5 min. An Agilent 6520 Q-ToF MS equipped with
a multimode ion source operating in electrospray ionization (ESI) negative (–) mode was used to
determine the chemical formula, molecular weight (MW), and quantity of each target compound.
All sample extracts were analyzed in full scan mode over 40–1000 Da. A mass accuracy of ± 10
ppm was selected for compound identification and quantification. Samples with individual NACs
exhibiting the highest MS signal intensities in full scan mode were re-examined in targeted MS-



MS mode using a collision-induced dissociation (CID) technique. The MS-MS spectra of target
NACs [M-H]⁻ ions were acquired to deduce structural information. Similar to bulk carbon and
light absorption measurements, NACs were primarily determined for CS- and HS-phase samples
with substantial OC loadings.

Due to the limited availability of authentic standards, many of the NACs identified in

cookstove combustion samples were quantified using surrogate compounds with similar MW or
structures. An internal standard method with a 9-point calibration curve (~0.01 – 2 ng µL) was
applied for quantification of concentrations. The compounds represented by each identified NAC
formula were quantified individually and combined to calculate the mass ratio of total NACs to
OC (µg m⁻³) × 100% (tNAC$_{OC}$%). Presently, the organic matter (OM) to OC ratio was not
measured or estimated for cookstove combustion emissions, so tNAC$_{OC}$% could be up to 2 times
greater than the contributions of NACs to OM (Reff et al., 2009; Turpin and Lim, 2001). Table S3
lists the chemical formulas, proposed structures, and standard assignments for the NACs identified
here. The quality assurance and control (QA/QC) procedures for filter extraction and instrumental
analysis were the same as Xie et al. (2017a, 2019). NACs were not detected in field blank and
background samples. The average recoveries of NAC standards on pre-spiked blank filters ranged
from 75.1% to 116%, and the method detection limit had a range of 0.70–17.6 pg.
***2.3. Data analysis***

In Xie et al. (2017a), the DAD measurement directly identified the chemical compounds in

chamber SOA responsible for light absorption in the near UV and visible light ranges. However,
no light absorption from individual NACs was detected in the DAD chromatograms from open BB
(Xie et al., 2019) and cookstove emissions (this work). So the contributions of individual NACs





to light absorption coefficient ($Abs_\lambda$, $Mm^{-1}$) for each sample extract at 365 nm ($Abs_{365,iNAC}\%$) were
calculated using the method described in Xie et al. (2017a, 2019):
$$Abs_{365,iNAC}\% = \frac{C_{iNAC} \times MAC_{365,iNAC}}{Abs_{365}} \times 100\% \qquad (1)$$
where $C_{iNAC}$ is the mass concentration (ng $m^{-3}$) of individual NACs, and $MAC_{365,iNAC}$ is the mass
absorption coefficient ($MAC_\lambda$, $m^2 g^{-1}$) of individual NACs at 365 nm. $Abs_{365}$ is the light absorption
coefficient ($Mm^{-1}$) of each sample extract at 365 nm, and has been widely used to represent BrC
absorption (Chen and Bond, 2010; Hecobian et al., 2010; Liu et al., 2013). Each NAC compound
was assumed to absorb as a standard (Table S3), of which the $MAC_{365,iNAC}$ value was obtained
from Xie et al. (2017a, 2019) and listed in Table S4. In this work, Student's *t*-test was used to
determine if the means of two sets of data are significantly different from each other, and a *p* value
less than 0.05 indicates significant difference.
**3 Results and discussion**
*3.1 Summary of total NACs concentration from cookstove emissions*

Table 1 summarizes the average concentrations of total NACs and average $tNAC_{OC}\%$ for

$Q_f$ and $Q_b$ by fuel type and WBT phase. Filter samples of emissions from burning red oak wood
show significantly ($p < 0.05$) higher average total NAC concentrations and $tNAC_{OC}\%$ than the
charcoal burning samples. Wood burning generates more volatile aromatic compounds (e.g.,
phenols, PAHs) than charcoal burning (Kim Oanh, et al., 1999), and NACs can form when
aromatic compounds and reactive nitrogen (e.g., $NO_X$) are present during solid fuel combustion
(Lin et al., 2016, 2017). While burning red oak, emissions from the CS and HS phases show similar
average NAC concentrations and $tNAC_{OC}\%$ (Table 1). Additionally, burning low or high moisture
red oak in the Jiko Poa stove shows no significant ($p > 0.05$) difference in $tNAC_{OC}\%$ (Tables S5
and S6). Thus, the NAC and OC emissions from red oak burning are less likely influenced by

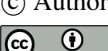



WBT phase or fuel moisture. For charcoal fuel samples, compared with the CS-phase, the HS-
phase shows significantly higher ($p < 0.05$) average NAC concentrations. This is likely due to the
increase in OC with the HS phase (Table 1), as the average $tNAC_{OC}\%$ values are much closer for
the CS- ($0.38 \pm 0.25\%$) and HS-phases ($0.31 \pm 0.22\%$).

Several studies have placed a quartz-fiber filter behind a PTFE filter to evaluate the positive

adsorption artifact — adsorption of gas-phase compounds onto particle filter media, "blow-on"
effect (Peters et al., 2000; Subramanian et al., 2004; Watson et al., 2009; Xie et al., 2014). This
method is expected to provide a consistent estimate irrespective of sampling time, but may over
correct the positive artifact by 16–20% due to volatilization of OC off the upstream PTFE filter
(negative artifact, "blow-off" effect) (Subramannian et al., 2004). The present study is the first to
consider sampling artifact when measuring semivolatile NACs. This concept merits consideration
as quantification of particle-phase NACs may be subject to large uncertainty. Table 1 shows that
the average concentrations of total NACs on $Q_b$ ($0.37 \pm 0.31 - 1.78 \pm 0.78\ \mu g\ m^{-3}$) are greater than
50% and 70% of those on $Q_f$ ($0.47 \pm 0.40 - 3.54 \pm 1.63\ \mu g\ m^{-3}$) for red oak and charcoal burning,
respectively. The average $Q_b$ to $Q_f$ ratio in percentage using OC concentrations is 2-3 times lower
($14.8 \pm 3.87 - 38.8 \pm 18.9\%$). Hence, the NACs identified in this work are present in the relatively
volatile bulk OC fraction emitted from cookstoves, and the NACs in the $Q_f$ samples may also be
present in the gas-phase in the atmosphere. Charcoal burning emissions show even higher ($p < 0.05$)
$Q_b$ to $Q_f$ total NAC mass ratios (CS $87.7 \pm 34.2\%$, HS $143 \pm 51.4\%$) than red oak burning (CS $53.0$
$\pm 10.6\%$, HS $55.1 \pm 24.7\%$), which is largely due to the higher OC loads on $Q_f$ samples from red
oak burning. Xie et al. (2018) assumed previously that the $Q_b$-adsorbed OC represented the
positive sampling artifact only, and adjusted the light absorbing properties of OC on $Q_f$ by
subtracting $Abs_{365}$ and OC of $Q_b$ samples directly. In this study, the high $Q_b$ to $Q_f$ ratios of total



NACs indicate that the volatilization of NACs from upstream PTFE filter cannot be neglected, but
the relative contributions of positive and negative artifacts to $Q_b$ measurements are unknown.
Therefore, the measurement results of NACs in $Q_f$ and $Q_b$ samples were provided separately, and
no correction was conducted for $Q_f$ measurements in this work. Since the gaseous NACs adsorbed
on $Q_b$ samples depends on $Q_f$ loadings, $tNAC_{OC}\%$ and total NACs concentrations in each $Q_f$-$Q_b$
pair from matching tests are significantly correlated ($p < 0.05$, Fig. S1a, b, d, and e).

Along with modified combustion efficiency (MCE), the EC/OC and BC/OA (organic

aerosol) ratios were used previously as indicators of biomass burning conditions (McMeeking et
al., 2014; Pokhrel et al., 2016). Here the burn condition indicates general flame intensity or
combustion temperature (Chen and Bond, 2010; Saleh et al., 2014), and is parameterized to
investigate combustion processes (e.g., pyrolysis). The MCE, EC/OC and BC/OA ratios are key
to understanding particulate OC absorptivity (Saleh et al., 2014; Lu et al., 2015) and NACs
formation from open BB (Xie et al., 2019). Presently, the relationships of $tNAC_{OC}\%$ versus EC/OC
for $Q_f$ samples are shown in Fig. S1c and f by fuel type. Because no significant difference was
observed for average total NACs concentrations, $tNAC_{OC}\%$, and EC/OC ratios when testing CS-
versus HS- phases during red oak fuel burning, the CS- and HS-phases were pooled for a regression
analysis. The $tNAC_{OC}\%$ of $Q_f$ samples positively correlate ($r = 0.83$, $p < 0.05$) with EC/OC for red
oak burning (Fig. S1c), as observed in Xie et al. (2019) for open BB, which suggests that burn
conditions influence NACs formation during BB. Note that the NAC concentrations on $Q_f$ were
possibly adsorbed while in a gaseous state, while EC is particle phase.

Like $MAC_{365}$ and $\mathring{A}_{abs}$ in $Q_f$ samples for charcoal burning (Xie et al., 2018), $tNAC_{OC}\%$

derived from the same samples did not correlate with EC/OC ratios in this work (Fig. S1f). Xie et
al. (2018) found that the HS-phase for charcoal burning had average OC EFs 5–10 times higher



than the CS-phase, while the EC EFs decreased by more than 90% from the CS- to HS-phase, so
the EC/OC for charcoal burning is sensitive to the initial temperature in the cookstove, and cannot
be used to predict burn conditions, BrC absorption, or NACs formation.
*3.2 Composition of NACs in $Q_f$ and $Q_b$*
During solid fuel combustion, NACs may form from aromatic compounds (e.g., substituted
phenols) and reactive nitrogen species (e.g., $NH_3$, $NO_X$, and HONO) in both the gas- and particle-
phase (Harrison et al., 2005; Kwamena and Abbatt, 2008; Lu et al., 2011; Lin et al., 2016, 2017).
Aromatic hydrocarbons are produced during fuel pyrolysis (Simoneit et al., 1993; Simoneit, 2002;
Kaal et al., 2009). Oxidation of fuel derived nitrogen, rather than molecular nitrogen in air, is the
major formation pathway of reactive nitrogen species (Glarborg et al., 2003).
Presently, seventeen chemical formulas were identified as NACs in cookstove emissions,
several of which are widely observed in ambient air and open BB particles (e.g., $C_6H_5NO_3$,
$C_6H_5NO_4$) (Claeys et al., 2012; Zhang et al., 2013; Lin et al., 2016, 2017; Xie et al., 2019). Figure
1 shows the average concentrations (ng m$^{-3}$) of individual NACs in $Q_f$ and $Q_b$ samples by fuel type
and WBT phase. The corresponding average mass ratios of individual NACs to OC $\times$ 100%
(iNAC$_{OC}$%) are shown in Fig. S2. Details of the NACs composition expressed in iNAC$_{OC}$% for
each fuel-cookstove experiment are given in Tables S5–S8.
Generally, the CS and HS phases have consistent NAC profiles for red oak combustion
(Figs. 1a, b and S2a, b). $C_{10}H_7NO_3$ has the highest average concentrations (CS-$Q_f$ 1003 $\pm$ 803 ng
m$^{-3}$, HS-$Q_f$ 1149 $\pm$ 1053 ng m$^{-3}$) and iNAC$_{OC}$% (CS-$Q_f$ 0.45 $\pm$ 0.80%, HS-$Q_f$ 0.43 $\pm$ 0.79%) on $Q_f$,
followed by $C_{11}H_9NO_3$, $C_{10}H_{11}NO_5$, and $C_{11}H_{13}NO_5$. $Q_b$ samples of red oak combustion emissions
have similar NACs profiles and characteristic species (e.g., $C_{10}H_7NO_3$, $C_{11}H_9NO_3$) as $Q_f$ samples,
and the individual NAC distributions in $Q_b$ to $Q_f$ samples are similar between the CS- and HS-





phases (Fig. 1a, b). It appears that the formation of NACs from red oak burning in cookstoves
depends largely on burn conditions reflected by EC/OC ratios (Fig. S1c) rather than WBT phases.
Among the 17 identified NACs from red oak burning, $C_8H_5NO_2$ and $C_{11}H_{13}NO_6$ have the lowest
$Q_b$ to $Q_f$ ratios (2.42 – 12.6%, Fig. 1a, b), indicating their low volatility. The low volatility of
$C_{11}H_{13}NO_6$ might be due to its relatively high MW; while $C_8H_5NO_2$ has the second lowest MW
and its structure likely contains functional groups that decrease vapor pressure (e.g., carboxyl
group) (Donahue et al., 2011).

Charcoal burning generated high abundances of $C_8H_9NO_5$, $C_{11}H_9NO_3$, and $C_{10}H_7NO_3$ for

both CS (86.6 ± 98.7 – 170 ± 200 ng m$^{-3}$) and HS (97.1 ± 38.5 – 178 ± 104 ng m$^{-3}$) phases (Figs.
1c, d and S2c, d). Average concentrations of $C_8H_9NO_5$, $C_{11}H_9NO_3$, and $C_{10}H_7NO_3$ in the $Q_b$ (62.0
± 64.9 – 198 ± 115 ng m$^{-3}$) and $Q_f$ samples were comparable. However, the iNAC$_{OC}$% of these
compounds are 1.45 ± 0.68 – 5.16 ± 2.84 times higher in $Q_b$ (iNAC$_{OC}$%, 0.11 ± 0.18 – 0.46 ±
0.69%) than in $Q_f$ samples (0.052 ± 0.067 – 0.14 ± 0.15%). High levels of $C_6H_5NO_4$, $C_7H_7NO_4$,
and $C_8H_9NO_4$ were also observed in the HS phase for charcoal burning (Fig. 1d). These compounds
in $Q_b$ samples had average concentrations (222 ± 132 – 297 ± 277 ng m$^{-3}$) 22.6 – 80.8% higher
than in $Q_f$ samples (150 ± 118 – 181 ± 111 ng m$^{-3}$). As such, the charcoal HS phase generates more
low MW NACs (e.g., $C_6H_5NO_4$, $C_7H_7NO_4$) than the CS phase, and the initial temperature in the
cookstove has an impact on NAC formation from charcoal burning.

As mentioned in section 3.1, using a $Q_b$ has been widely applied to evaluate the positive

sampling artifact for OC and semivolatile organic compounds. This method might only work for
bulk PM, OC, and low volatile organic compounds, of which the concentrations in $Q_b$ samples are
much lower than $Q_f$ samples and usually presumed to be due to positive adsorption artifacts only
(Subramanian et al., 2004; Watson et al., 2009). In this work, the average $Q_b$ to $Q_f$ mass ratios of





the 17 individual NACs ranged from 54.3 ± 24.5% to 135 ± 52.4%, and the evaporation of NACs
from $Q_f$ (negative artifact) is unknown. As a result, the particle-phase NAC concentrations cannot
be calculated by simply subtracting $Q_b$ measurements from those of $Q_f$. Considering that most of
the $Q_f$ and $Q_b$ samples were collected near ambient temperature (Table S2, ~25 $^o$C), some of the
identified NACs (e.g., 4-nitrophenol) may have substantial gas-phase concentrations (Li et al.,
2020), and the composition of NACs derived from $Q_f$ measurements alone can be biased due to
the lack of gas-phase measurements. Future work is needed to evaluate the composition of NACs
from emission sources in both the particle and gas phases.
***3.3 Identification of NACs structures***

Figures S3 and S4 exhibited extracted ion chromatograms (EICs) and MS-MS spectra of

the 17 identified NACs. For comparison, the MS-MS spectra of standard compounds used in this
work are obtained from Xie et al. (2017a, 2019) and shown in Fig. S5. Among all identified NAC
formulas, $C_{10}H_7NO_3$ was detected in each fuel-cookstove experiment (Tables S5 – S8) and showed
the highest concentrations in emissions from burning red oak (Fig. 1a, b). The MS-MS spectrum
of $C_{10}H_7NO_3$ (Fig. S4l) is like 2-nitro-1-phenol (Fig. S5g) but shows a ~1 min difference in
retention time (Fig. S3i 10.9 min, 2-nitro-1-phenol 11.8 min). $C_{10}H_7NO_3$ is presumed to be an
isomer of 2-nitro-1-phenol with a nitronaphthol structure. $C_{11}H_9NO_3$ has a degree of unsaturation
and a fragmentation pattern (Fig. S4q) like $C_{10}H_7NO_3$ and is likely a structural isomer of methyl
nitronaphthol. $C_6H_5NO_3$, $C_7H_7NO_3$, $C_6H_5NO_4$, and $C_7H_7NO_4$ are commonly detected in
combustion emissions (Lin et al., 2016, 2017; Xie et al., 2017a) and atmospheric particles (Claeys
et al., 2012; Zhang et al., 2013). $C_6H_5NO_3$ and $C_6H_5NO_4$ are identified as 4-nitrophenol and 4-
nitrocatechol using authentic standards (Figs. S4a, d and S5a, c). $C_7H_7NO_3$ has two isomers (Fig.
S3b) and the compound eluting at 9.98 min has the same retention time and MS-MS spectrum (Fig.



S4c) as 2-methyl-4-nitrophenol (Fig. S5b). In ambient PM and chamber SOA, $C_7H_7NO_4$ was
identified using standard compounds as a series of methyl-nitrocatechol isomers (4-methyl-5-
nitrocatechol, 3-methyl-5-nitrocatechol, and 3-methyl-6-nitrocatechol) (Iinuma et al., 2010).
According to the HPLC-Q-ToFMS data for $C_7H_7NO_4$ identified in Iinuma et al. (2010) and our
previous studies (Xie et al., 2017a, 2019), the two $C_7H_7NO_4$ isomers in Fig. S3d are likely 4-
methyl-5-nitrocatechol and 3-methyl-6-nitrocatechol, respectively. Here we cannot rule out the
presence of 3-methyl-5-nitrocatechol, which may co-elute with 4-methyl-5-nitrocatechol (Iinuma
et al., 2010). In Fig. S4k, o, and p, the MS-MS spectra of $C_7H_7NO_5$, $C_8H_7NO_5$, and $C_8H_9NO_5$ all
show a loss of $CH_3$ + NO (or $NO_2$) + CO. The loss of $CH_3$ is typically due to a methoxy group in
NAC molecules, and NO (or $NO_2$) and CO loss is commonly observed for NACs with more than
one phenoxy group (Xie et al., 2019). So methoxy nitrophenol is the proposed skeleton for
$C_7H_7NO_5$, $C_8H_7NO_5$, and $C_8H_9NO_5$. Other functional groups were estimated using their chemical
formulas and degree of unsaturation as a basis (Table S3).

The present study quantifies $C_8H_7NO_4$ and $C_9H_9NO_4$ using 2-methyl-5-benzoic acid

($C_8H_7NO_4$) and 2,5-dimethyl-4-nitrobenzoic acid ($C_9H_9NO_4$), respectively. The fragmentation
patterns of $C_8H_7NO_4$ (Fig. S4g, h) and $C_9H_9NO_4$ compounds (Fig. S4m, n) are different from their
corresponding surrogates (Fig. S5f, h) and loss of $CO_2$ is not observed, so $C_8H_7NO_4$ and $C_9H_9NO_4$
compound structures do not include a carboxyl group. The MS-MS spectra of $C_8H_7NO_4$ eluting at
8.14 min (Fig. S3e) and $C_9H_9NO_4$ eluting at 9.22 min (Fig. S3j) indicate the loss of OCN (Fig.
S4g, m), suggesting benzoxazole/benzisoxazole structure or the presence of cyanate ($-O-C\equiv N$) or
isocyanate ($-O=C=N$) groups. Mass spectra of selected standard compounds (Fig. S5i-n) in our
previous work (Xie et al. 2019) show the loss of an OCN group only happens during the
fragmentation of phenyl cyanate. Thus, the $C_8H_7NO_4$ and $C_9H_9NO_4$ isomers containing OCN





indicate a phenyl cyanate feature. However, the fragmentation mechanism related to the loss of a
single nitrogen for the second $C_8H_7NO_4$ isomer (Fig. S3e, Fig. S4h) is unknown and requires
further study. The MS-MS spectrum of the second $C_9H_9NO_4$ isomer had dominant ions at $m/z$ 194
($[M–H]^-$), 164 (loss of NO), and 149 (loss of NO + $CH_3$). Compared with the MS-MS spectra of
4-nitrophenol and 2-methyl-4-nitrophenol (Fig. S5a, b), the second $C_9H_9NO_4$ isomer is likely a
methoxy nitrophenol with an extra ethyl group.

The EIC signal of $C_8H_9NO_4$ in Fig. S3f comprises at least 3-4 isomers, and the MS-MS

spectra are always dominated by ions at $m/z$ 182 ($[M–H]^-$), 152 (loss of NO), and 137 (loss of NO
+ $CH_3$) with some changes in relative abundance. The fragmentation mechanism of $C_8H_9NO_4$
represented by the MS-MS spectrum in Fig. S4i is consistent with that of the second $C_9H_9NO_4$
isomer (Fig. S4n), so the $C_8H_9NO_4$ might also have a methoxy nitrophenol skeleton.

Unlike other NACs, $C_8H_5NO_2$ was only detected in samples from red oak burning in the

Jiko Poa and charcoal burning in the Éclair (Tables S5 – S8). The average mass ratios of $C_8H_5NO_2$
in $Q_b$ to $Q_f$ samples for red oak burning are less than 15% (CS phase 2.42%, HS phase 12.6%),
and $C_8H_5NO_2$ was not detected in any $Q_b$ samples for charcoal burning. The MS-MS spectrum of
$C_8H_5NO_2$ is characterized by $CO_2$ loss (Fig. S4j), indicative of a carboxyl group. Considering the
degree of unsaturation of the $C_8H_5NO_2$ molecule and the cyano group feature in BB tracers (e.g.,
hydrogen cyanide, benzonitrile) (Schneider et al., 1997; Li et al., 2000; Gilman et al., 2015),
$C_8H_5NO_2$ may be a cyanobenzoic acid.

The $C_{10}H_{11}NO_4$, $C_{10}H_{11}NO_5$, $C_{11}H_{13}NO_5$, and $C_{11}H_{13}NO_6$ detected here are also observed

in other BB experiments (Xie et al., 2019). Their MS-MS spectra are characterized by the loss of
at least one $CH_3$ and/or OCN (Fig. S4r–u), suggestive of methoxy or cyanate groups. Without





authentic standards, fragmentation patterns (Fig. S4r–u) were used to determine the molecular
structures of $C_{10}H_{11}NO_4$, $C_{10}H_{11}NO_5$, $C_{11}H_{13}NO_5$, and $C_{11}H_{13}NO_6$ (Table S3).

Nearly all NAC formulas identified in this work were observed previously (Lin et al., 2016,

2017; Xie et al., 2017a; Fleming et al., 2018; Xie et al., 2019). Few studies attempt to retrieve
structural information for NACs using MS-MS spectra of authentic standards. Although multiple
NACs may be generated from BB and photooxidation of aromatics in the presence of $NO_X$, NAC
structures may differ across emission sources. Xie et al. (2019) found that fragmentation patterns
of $C_7H_7NO_5$ and $C_8H_9NO_5$ from BB and photochemical reactions are distinct, and the methoxy
and cyanate groups are featured only in BB NACs. Thus, knowing the NAC structure is useful to
emissions source identification. In this work, the chemical and structural information obtained for
NACs sampled during red oak and charcoal burning are similar, presumably because the charcoal
fuel used is produced by the slow pyrolysis of wood. However, NACs in red oak and charcoal
burning emissions can be differentiated compositionally. As shown in Figs. 1 and S2, the NAC
emissions from red oak burning in cookstoves are characterized by $C_{10}H_7NO_3$ and $C_{11}H_9NO_3$. In
addition to these two species, charcoal burning in cookstoves also generates high fractions of
$C_8H_9NO_5$ (Fig. S2c, d). This difference among NACs may help with source apportionment. Figure
2 compares NAC composition from cookstove emissions, open BB (Xie et al., 2019), and SOA
chamber experiments (Xie et al., 2017a). Since previous source emissions studies ignored $Q_b$
measurements and normalized individual NACs concentrations to OM, only $Q_f$ measurements in
this work are compared (Fig. 2a, b) with their iNAC$_{OC}$% values multiplied by 1.7 (proposed
OM/OC ratio, Reff et al., 2009). The three open BB tests (Fig. 2c) were conducted with two fuel
types under different ambient temperatures (10–29 °C) and RH% (49–83%) (Xie et al., 2019). But
they consistently emit $C_6H_5NO_4$, $C_7H_7NO_4$, and $C_9H_9NO_4$, which is compositionally distinct from





cookstove emissions (Fig. 2a, b). Moreover, the average mass contribution of total NACs to OM
for open BB ($0.12 \pm 0.051\%$) was 4–14 times lower than that for cookstove emissions. This result
is likely due to the high temperature flaming combustion produced in the cookstoves (Shen et al.,
2012; Xie et al., 2018). In Fig. 2d and e, the NAC profiles yielded for photochemical reactions
appear to have aromatic precursors. Therefore, the source of NACs can be identified by combining
their characteristic structures and composition. The filter-based NACs reported for the experiments
shown in Figure 2 were all measured using the identical method and HPLC-Q-ToFMS instrument,
reducing any potential methodological bias. However, total gas-phase NAC concentrations need
to be properly sampled and measured to account for the impact of gas/particle partitioning on their
distribution.
***3.4 Contributions of NACs to Abs$_{365}$***
The average Abs$_{365,iNAC}\%$ values of $Q_f$ and $Q_b$ samples are presented by fuel type and WBT
phase in the Fig. 3 stack plots, and experimental data for each fuel-cookstove are provided in
Tables S9–S12. The average contributions of total NACs to Abs$_{365}$ (Abs$_{365,tNAC}\%$) of the sample
extracts ($Q_f$ $1.10 - 2.58\%$, $Q_b$ $10.7 - 21.0\%$) are up to 10 times greater than their average tNAC$_{OC}\%$
($Q_f$ $0.31 - 0.97\%$, $Q_b$ $1.08 - 3.31\%$, Table 1). Considering that some NACs are not light-absorbing
(Table S4) and the OM/OC ratio is typically greater than unity, most NACs that contribute to
Abs$_{365}$ are strong BrC chromophores. Like the mass composition of NACs (Fig. 1), $C_{10}H_7NO_3$ (CS
0.24%, HS 0.43%) and $C_8H_9NO_5$ (CS 1.22%, HS 0.55%) were the major contributors to Abs$_{365}$
for the $Q_f$ samples collected during red oak and charcoal burning, respectively (Fig.3a). However,
the identified NACs only explain a minor fraction ($< 5\%$) of bulk extract absorption. Chen and
Bond (2010) hypothesized that BrC absorption is strongly associated with large molecules
containing conjugated aromatic rings and functional groups. Additionally, Di Lorenzo et al. (2017)





demonstrated that the majority of BrC absorption arises from large molecules with MW > 500–
1000 Da. In previous studies, less than 10% of the BrC absorption at $\lambda = 365$ nm from ambient or
BB particles are ascribed to NACs with MW < 300 Da. Further studies are needed to identify these
larger molecules that are the dominant light absorbers in BB and cookstove PM. The average
$Abs_{365,tNAC}$% of $Q_b$ samples are 7.52 to 11.3 times higher than those of $Q_f$ samples. Unlike the $Q_f$
samples from red oak burning, $C_{10}H_{11}NO_5$ (CS 2.77%, HS 3.09%) has the highest average
contribution to $Abs_{365}$ for $Q_b$ samples, followed by $C_{10}H_7NO_3$ (CS 1.96%, HS 1.32%) and
$C_8H_9NO_5$ (CS 1.32%, HS 1.44%). While $C_8H_9NO_5$ dominated the contribution (CS 8.78%, HS
5.82%) to $Abs_{365}$ for the $Q_b$ samples from charcoal burning (Fig. 3b). As mentioned in section 3.2,
some NACs identified in this work might have substantial gas-phase concentrations. Jacobson
(1999) inferred that the nitrate-bearing aromatic gases may play a role in reducing the UV
irradiance within the boundary layer in Los Angles during 1973 – 1987. Therefore, we suspect that
gaseous NACs may be an important group of molecules absorbing in short UV region in the
atmosphere.
**4 Conclusion**
This study investigated the composition, chemical formulas, and structures of NACs in
$PM_{2.5}$ emitted from burning red oak and charcoal in a variety of cookstoves. Total NAC mass and
compositional differences between $Q_f$ and $Q_b$ samples suggest that the identified NACs have
substantial gas-phase concentrations. By comparing the MS-MS spectra of identified NACs to
standard compound spectra, the structures of NACs featuring methoxy and cyanate groups in
cookstove emissions are confirmed. The source identification of NACs would be less ambiguous
if both the structures and composition of NACs are known, as different emission sources have
distinct NAC characteristics. Similar to previous work, the average contribution of total NACs to



Abs$_{365}$ of Q$_f$ samples is less than 5% (1.10 – 2.58%), suggesting the need to shift our focus from
NACs (MW < 300 Da) to the chemical and optical properties of large molecules (e.g., MW > 500
Da) in particles. However, their average contributions to Abs$_{365}$ of Q$_b$ samples are 7.52 to 11.3
times higher, so NACs may be important light absorbers in the gas phase. Further research in
understanding the influence of gaseous chromophores on the earth's radiative balance is warranted.

*Data availability*
Data used in the writing of this manuscript is available at the U.S. Environmental Protection
Agency's Environmental Dataset Gateway (https://edg.epa.gov).

*Competing interests*
The authors declare that they have no conflict of interest.

*Disclaimer*
The views expressed in this article are those of the authors and do not necessarily represent the
views or policies of the U.S. Environmental Protection Agency.

*Author contribution*
MX and AH designed the research. MX, ZZ, and XC performed the experiments. GS and JJ
managed sample collection. MX and MH analyzed the data and wrote the paper with significant
contributions from AH and QW.

*Acknowledgements*



This research was supported by the National Natural Science Foundation of China (NSFC,
41701551), the Startup Foundation for Introducing Talent of NUIST (No. 2243141801001), and
in part by an appointment to the Postdoctoral Research Program at the Office of Research and
Development by the Oak Ridge institute for Science and Education through Interagency
Agreement No. 92433001 between the U.S. Department of Energy and the U.S. Environmental
Protection Agency. We thank B. Patel for assistance on ECOC analysis of $PM_{2.5}$ filters.

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





Table 1. Average concentrations of total NACs and tNAC$_{OC}$% in Q$_f$ and Q$_b$ samples by fuel type and WBT phase.

| Fuel &Test phase | Red Oak | | Charcoal | |
|---|---|---|---|---|
| | CS | HS[a] | CS | HS |
| *Front filter (Q$_f$)* | | | | |
| Sample number | 18 | 17[b] | 15 | 15 |
| total NAC (μg m$^{-3}$) | 3.22 ± 1.36 | 3.54 ± 1.63 | 0.47 ± 0.40 | 0.97 ± 0.46 |
| tNAC$_{OC}$% | 0.97 ± 1.07 | 0.94 ± 1.10 | 0.38 ± 0.25 | 0.31 ± 0.22 |
| OC (μg m$^{-3}$)[c] | 624 ± 410 | 908 ± 885 | 115 ± 72.0 | 447 ± 271 |
| EC/OC[c] | 1.74 ± 1.42 | 1.96 ± 1.74 | 6.12 ± 2.76 | 0.029 ± 0.012 |
| *Backup filter (Q$_b$)* | | | | |
| Sample number | 18 | 17[b] | 14[b] | 15 |
| total NAC (μg m$^{-3}$) | 1.67 ± 0.76 | 1.78 ± 0.78 | 0.37 ± 0.31 | 1.30 ± 0.70 |
| tNAC$_{OC}$% | 3.31 ± 3.46 | 2.76 ± 2.67 | 1.10 ± 0.89 | 1.08 ± 0.51 |
| OC (μg m$^{-3}$)[c] | 78.4 ± 43.2 | 100 ± 58.4 | 41.9 ± 23.3 | 138 ± 70.8 |
| *Q$_b$/Q$_f$ ratio (%)* | | | | |
| total NACs | 53.0 ± 10.6 | 55.1 ± 24.7 | 87.7 ± 34.2 | 143 ± 51.4 |
| OC[c] | 14.8 ± 3.87 | 15.3 ± 6.37 | 35.4 ± 12.2 | 38.8 ± 18.9 |

[a] Including three SIM phase samples from the 3-stone fire; [b] one filter sample was missed for analysis; [c] data were obtained from Xie et al. (2018).



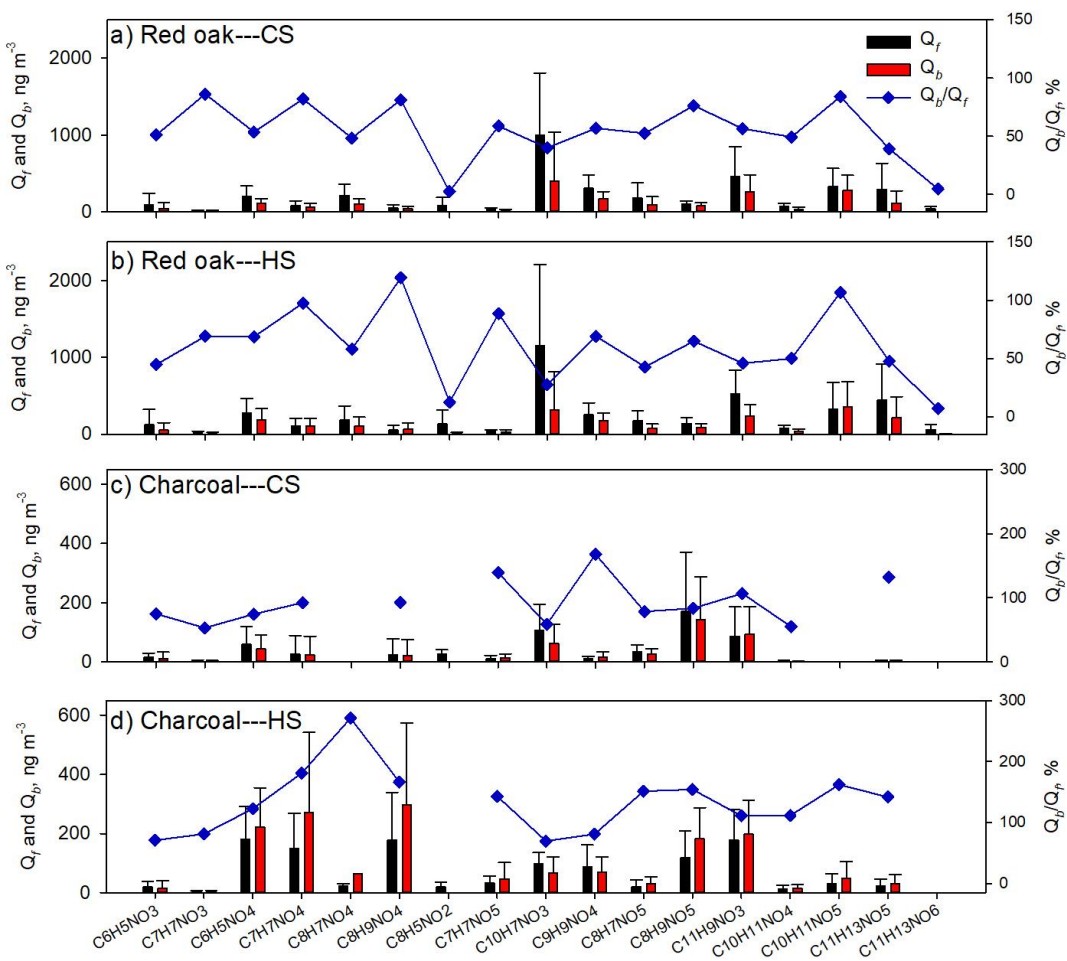

Figure 1. Average concentrations of individual NACs in $Q_f$ and $Q_b$ samples for (a) red oak burning under the CS phase, (b) red oak burning under the HS phase, (c) charcoal burning under the CS phase, and (d) charcoal burning under the HS phase. The blue scatters in each plot are mass ratios of individual NACs in $Q_b$ to $Q_f$ samples × 100%.



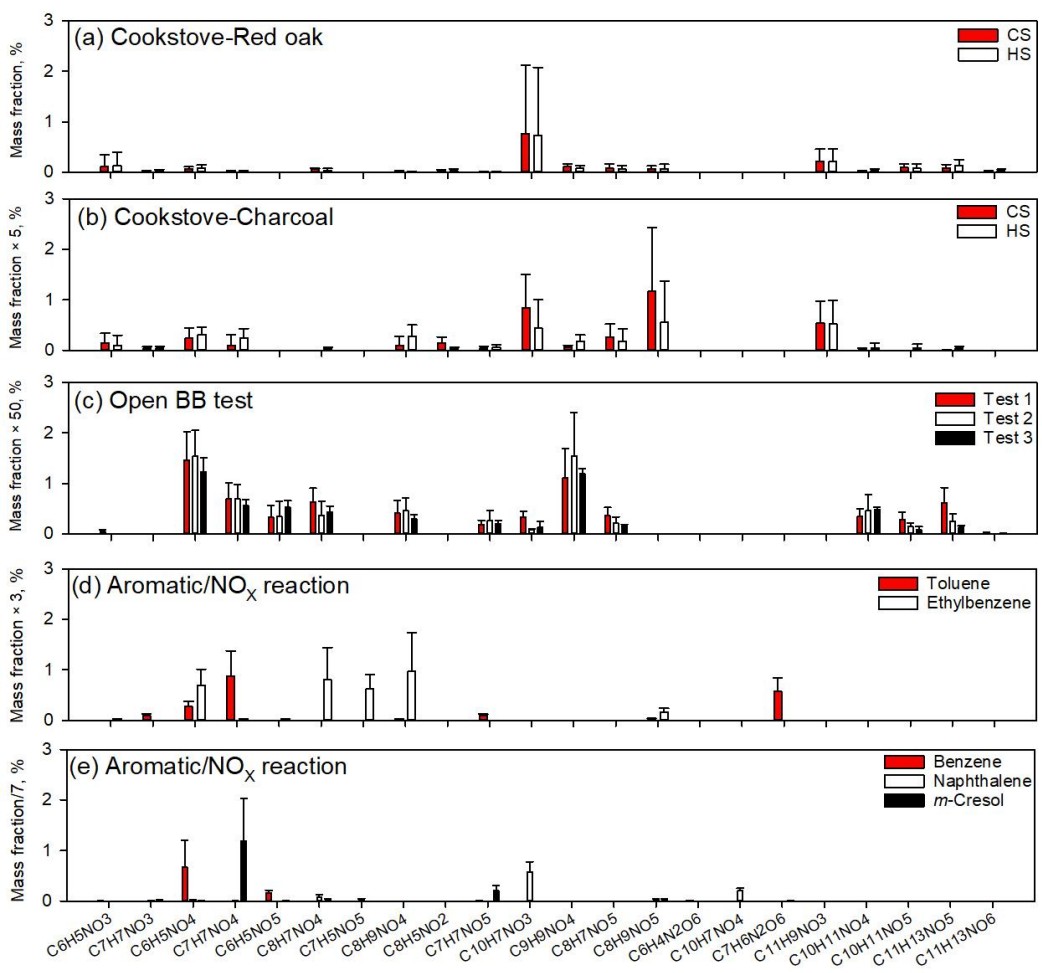

Figure 2. Average mass ratios (%) of individual NACs to organic matter from (a) red oak burning in cookstoves, (b) charcoal burning in cookstoves, (c) open BB experiments (Xie et al., 2019), photochemical reactions of (d) toluene and ehtylbenzene, and (e) benzene, naphthalene, and m-cresol with $NO_X$ (Xie et al., 2017a).





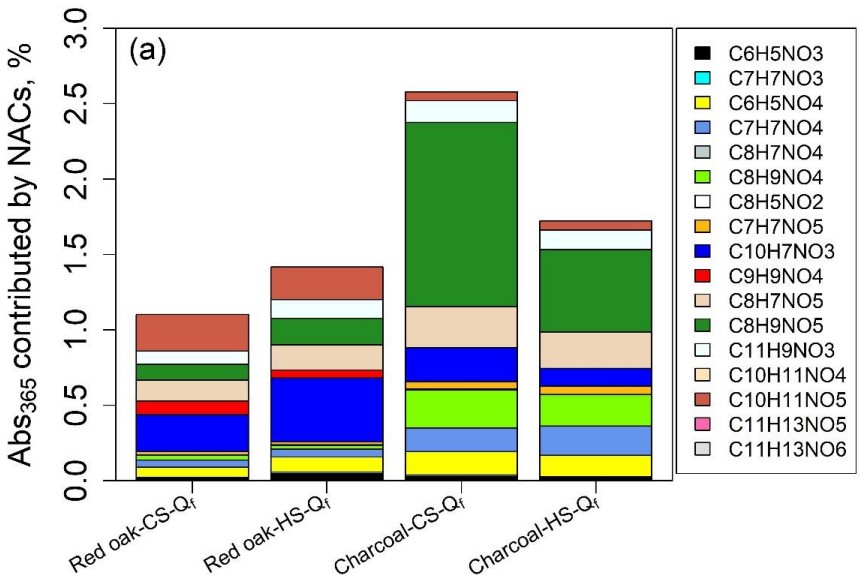

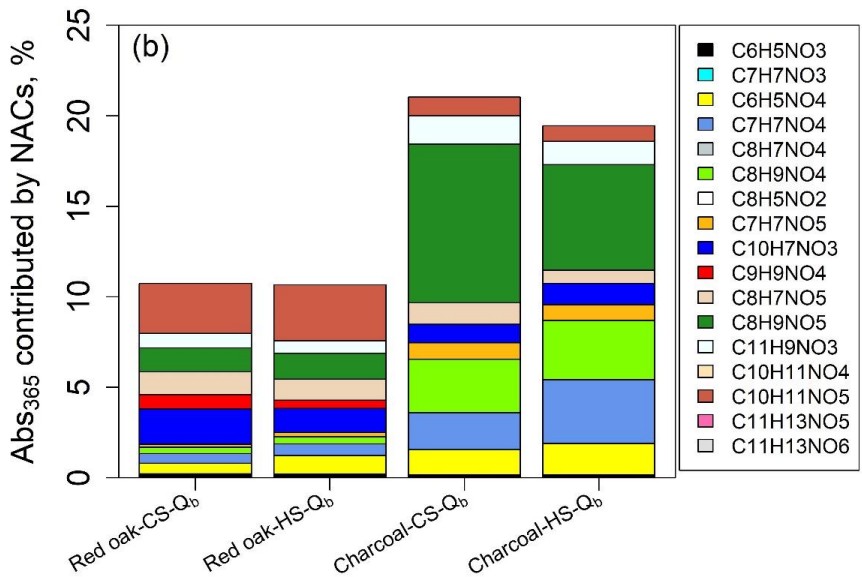

Figure 3. Average contributions (%) of individual NACs to bulk extracts Abs$_{365}$ of (a) Q$_f$, and (b) Q$_b$ samples from burning red oak and charcoal in cookstoves under CS and HS phases.