# Peer review of "Chemical composition, structures, and light absorption of N-containing 1 aromatic compounds emitted from burning wood and charcoal in 2 household cookstoves 3 Mingjie Xie1, Zhenzhen Zhao1, Amara L. Holder2, Michael D. Hays2, Xi Chen2, Guofeng Sh"

_Atmospheric Chemistry and Physics, 2020_

## Referee Comment (RC1) · Anonymous Referee #1 · 17 Jul 2020

Xie et al. identified and quantified individual nitrogen-containing aromatic compounds (NACs) found in cookstove aerosol produced from water boiling tests. The study focused on two different fuels, charcoal and red oak, and mainly compared and contrasted emissions of NACs from cold start and hot start phases of the WBT. A unique aspect of this study is a focus on filter artifacts by comparing NACs on a quartz fiber filter placed downline of a PTFE membrane. In addition, they quantified the absorption of individual NACs at 365 nm based on their measured concentrations. The authors identified 17 different structures of NACs from their MS-MS spectra. The main con-

clusions of this study are that the back up quartz fiber filter concentrations of NACs were very high, sometimes even larger than on the front PTFE filter highlighting the importance of understanding these sampling artifacts for quantification of semivolatile species better. They also conclude that the NACs in this study make up less than 5% of the extractable absorption at 365 nm on the PTFE filter.

General comments: The results of this paper should be published because this study quantifies particulate emissions of NACs from cookstoves, which is understudied. The results also demonstrate the need to understand sampling artifacts from filters when they are used for quantitative analysis. However, some of the key conclusions of the paper may be misleading for the reader. For example, it is concluded that <5% of the extractable absorption is from NACs and they not significant brown carbon chromophores in cookstove smoke. However, much higher percentages were observed on the back up quartz filter, some of which may be in the particle phase in the atmosphere.

Specific comments:

1. There are some well-documented problems with WBTs, mostly arguing that their combustion efficiencies don't match those in the real world (Johnson et al., 2008, 2010). If the combustion efficiency in real homes is lower, this could result in less NACs due to less flaming and lower NOx. Given this, it would be helpful to have a measure of combustion efficiency, such as modified combustion efficiency, so that it can be compared with field measurements in the future. This may be possible, given the paper mentions gaseous pollutants were measured (Line 140). Even without this, it would be helpful to have more of a description of the cookstoves and WBTs which would help with the interpretation of the results.

1a. The stoves are listed in tables in the supplement, however, they are not really discussed in the experimental section of the main paper. How are they different? Where are they used around the world?

1b. Please include more information about the water boiling tests in the experimental

section, as most readers of the journal will not be familiar with it. You should also mention the simmer phase is included for the hot start sample in some tests, if this is correct.

1c. Could use more reasoning as to how red oak and charcoal are different as seen in Figure S1 C and F by relating hot start and cold start phases to the observed types of combustion. For example, hot start is mostly smoldering for charcoal (high OC emissions with very low BC and therefore low NOx to make less NACs).

2. Regarding source apportionment for NAC measurements (Lines 385-402), these fractions of NAC/OM will be very different in the field because OM can come from many sources. The NAC should be ratioed to a combustion product such as CO or EC.

3. It is implied in lines 412-420 that NACs identified in this study are not significant BrC chromophores, however, if the quartz filter (Qb) is included the fraction is likely higher. It may be more appropriate to give an upper limit given that NACs on Qb could partition into the particle phase in the atmosphere. It is difficult to conclude that NACs are not significant BrC chromophores given the measurements on the sampling artifact that other studies have not considered. Also, NACs may be higher for fuel/stove/cooking activity combinations that result in more flaming combustion which produces NOx, an important reactant for NAC formation. Another factor is that the fractional absorption by NACs was not directly measured. Surrogates were used to quantify NAC concentrations and approximate MACs were used to calculate the Abs365.

4. It is assumed that because Abs365,tNAC% at 365 nm is 7-11 times higher on the quartz fiber backup filter, that NACs may be important light absorbers in the gas phase (lines 442, 425-429, 432-434). To claim this in the paper, more discussion and reasoning for should be given.

4a. Those on the backing filter are not necessarily in the gas phase in the natural environment. As you explain in the paper, there are both positive and negative artifacts and there is not likely a straightforward way of calculating what would be in the gas

phase.

4b. The vapor pressures of these molecules are very low, and the fraction in the gas phase is low. However, for some nitroaromatics such as 2-nitropenol the vapor pressure is higher. Are the concentrations for some molecules higher on the back up filter compared to the front filter and do we expect them to have higher vapor pressures?

4c. What are the absorption cross sections for these molecules in the gas phase and their expected gaseous concentrations that would lead us to believe they are significant? Are they long-lived enough in the gas phase to be important? Only solution phase MACs at 365 nm are used to claim that gas phase absorption is significant and this is not sufficient.

5. Line 132: Omit that you did kerosene tests. It is not critical as you do not discuss these results.

References

Johnson, M., Edwards, R., Alatorre Frenk, C. and Masera, O.: In-field greenhouse gas emissions from cookstoves in rural Mexican households, Atmos. Environ., 42(6), 1206–1222, doi:10.1016/j.atmosenv.2007.10.034, 2008.

Johnson, M., Edwards, R., Berrueta, V. and Masera, O.: New Approaches to Performance Testing of Improved Cookstoves, Environ. Sci. Technol., 44(1), 368–374, doi:10.1021/es9013294, 2010.

---

## Referee Comment (RC2) · Anonymous Referee #2 · 29 Jul 2020

This manuscript presents the analysis of N-containing aromatic compounds (NAC) in PM2.5 samples collected from biomass-burning emissions of wood and charcoal in special household stoves. Prior to the HPLC analysis, the collected filters were spiked with one deuterated internal standard and extracted in methanol. The goal of this research was to estimate the contribution of BrC NAC species to the total absorption of PM2.5 samples. The authors also discussed the differences (in OC, total NAC, individual NACs, etc.) between the hot-start and cold-start phases, and also between front and back filters. The authors acknowledged the limitations of this study (e.g., no

gas-phase NACs were measured). This study is scientifically important, since NACs are not only light-absorbing compounds, but also are toxic organic species and they are still not well characterized. The manuscript is well organized and well written. I have a few major comments:

Major comments:

1. The filter samples were spiked with only one deuterated I.S. compound (4-nitrophenol-d4, C6), while the analyzed NACs (Fig 1) have different volatility levels (C7-C11). The author should check if there were potential losses of I.S., which is more volatile than the rest of the analyzed species, and if these losses led to a large overestimates of the concentrations of the analyzed compounds.

2. The manuscript contains a lot of abbreviations, which made it very hard to read (HS, CS, Qf, Qb, WBT, OMMs, SIM, etc.)

3. Table 1. The concentrations of the total NACs are strikingly high for the backup filters. I am wondering if some sort of unexpected breakthrough happened during the sampling (especially in the case of charcoal burning). Would it be possible that the BB emissions were quite hot during the sampling, which caused the evaporation from the front filter?

Some minor comments Line 130. U.S. EPA – please make sure abbreviations are explained in the text Line 131: "USA" should be added after "NC" Line 126, what is "Jiko Poa"? Should company name be added? Lines 152, 156 etc. Company name (+city, state, country) of material and instruments is missing.

In summary, I recommend this manuscript for publication after major revisions

---

## Referee Comment (RC3) · Anonymous Referee #3 · 17 Aug 2020

General comments:

This manuscript is a nice piece of work describing the emission of nitrogen containing aromatic compounds (NAC) from the use of cookstoves. The authors aim to understand the contribution of these species to the light absorption of organic matter in PM2.5 at the wavelength of 365 nm. The authors found that much higher contribution of NAC light absorption to PM2.5 in quartz fiber backup filters than in PTFE front filters, suggesting that NAC may be an important group of light absorbing compounds in the gas phase. In addition, the authors found the NAC compounds targeted in this
manuscript (Mw < 300) are less important for the light absorption of PM2.5 bound organic matter at least from cookstove emissions, and indicate that larger molecules with Mw > 500 are responsible for the light absorption of organic matters. The manuscript is well written, and I recommend publication of this manuscript after addressing minor technical corrections outlined below.

Specific comments:

If it is possible, the authors should present all NAC in emission factors (g kg-3) rather than in mass concentrations ($\mu$g m$^{-3}$). Emission factors are more useful than mass concentrations as they can be used for emission control strategies directly.

Line 171: ng $\mu$L should be ng/$\mu$L or ng $\mu$L$^{-1}$.

Line 214 onwards: This assumes that the backup quartz fiber filter can trap all the gas phase compounds and does not have a breakthrough at all. A better way to estimate the artifact of a filter sampling system is to utilize a denuder in front of a PTFE filter for gas sampling and place a quartz fiber filter after the PTFE to correct a negative artifact from blown-off. It would be good if the authors discuss briefly here about the potential usage of the denuder for the artifact correction in addition to the backup quartz fiber filter.

Line 283: In Table S3, the authors suggest $C_8H_5NO_2$ as 3-cyanobenzoic acid. This is commercially available from several chemical suppliers, and it can be positively identified if $C_8H_5NO_2$ correspond to the authors' suggestion. I ask the authors to purchase the standard compound and quantify it instead of using a surrogate compound.

Fig S5: The mass spectrometric conditions and ionization methods used to obtain MS2 spectra for some compounds shown in Fig S5 are very different from ones shown in Fig S4, and they are not comparable at all. I ask the authors to remove those (EI and ESI+) from Fig S5 as they cannot be compared to each other.

[Figure]

2020.

---

## Author Comment (AC1) · 19 Sep 2020

**Reviewer 1**

Xie et al. identified and quantified individual nitrogen-containing aromatic compounds (NACs) found in cookstove aerosol produced from water boiling tests. The study focused on two different fuels, charcoal and red oak, and mainly compared and contrasted emissions of NACs from cold start and hot start phases of the WBT. A unique aspect of this study is a focus on filter artifacts by comparing NACs on a quartz fiber filter placed downline of a PTFE membrane. In addition, they quantified the absorption of individual NACs at 365 nm based on their measured concentrations. The authors identified 17 different structures of NACs from their MS-MS spectra. The main conclusions of this study are that the backup quartz fiber filter concentrations of NACs were very high, sometimes even larger than on the front PTFE filter highlighting the importance of understanding these sampling artifacts for quantification of semivolatile species better. They also conclude that the NACs in this study make up less than 5% of the extractable absorption at 365 nm on the PTFE filter.

General comments: The results of this paper should be published because this study quantifies particulate emissions of NACs from cookstoves, which is understudied. The results also demonstrate the need to understand sampling artifacts from filters when they are used for quantitative analysis. However, some of the key conclusions of the paper may be misleading for the reader. For example, it is concluded that <5% of the extractable absorption is from NACs and they not significant brown carbon chromophores in cookstove smoke. However, much higher percentages were observed on the back up quartz filter, some of which may be in the particle phase in the atmosphere.

*Reply:*

Thanks for the reviewer's comments, and we'll reply these point by point in the reviewer's specific comments.

Here we just want to clarify that NACs were analyzed only for quartz filter samples (front and backup filters, $Q_f$ and $Q_b$). PTFE filters were commonly used for gravimetric analysis, but were rarely extracted using organic solvents. The installation of a backup quartz filter ($Q_b$) behind a PTFE filter in parallel to a bare quartz filter ($Q_f$) was typically used to estimate the adsorption of gaseous OC ("positive artifact") on the main (or "bare") quartz filter (Subramanian et al., 2004; Watson et al., 2009). This Q-QBT approach presumes that the upstream PTFE filter adsorb no organic gases, and then $Q_b$ is exposed to organic vapor with the same concentration as $Q_f$. It has been shown to provide a robust estimate of the positive artifact on $Q_f$ OC (McDow and Huntzicker, 1990; Turpin et al., 1994).

In section 2.2 (lines 166-167), we have mentioned that the $Q_f$ and $Q_b$ sample extraction and subsequent analysis for NACs were conducted as described in Xie et al. (2019).

To avoid the confusion, the original expressions

"*to evaluate the potential for sampling artifacts of NACs in PM$_{2.5}$.*" (lines 120-121)

"*Adsorption artifact was evaluated using a quartz-fiber back-up filter ($Q_b$) installed downstream of the PTFE filter during PM$_{2.5}$ sampling.*" (lines 143-144)

have been changed into

"*to evaluate the potential for sampling artifacts of PM$_{2.5}$ NACs on the bare quartz filter in parallel.*" (lines 119-120)

"*The adsorption artifact of Q$_f$ was evaluated using a quartz-fiber back-up filter (Q$_b$) installed downstream of the PTFE filter during PM$_{2.5}$ sampling.*" (lines 147-148)

**Specific comments:**
**1.** There are some well-documented problems with WBTs, mostly arguing that their combustion efficiencies don't match those in the real world (Johnson et al., 2008, 2010). If the combustion efficiency in real homes is lower, this could result in less NACs due to less flaming and lower NOx. Given this, it would be helpful to have a measure of combustion efficiency, such as modified combustion efficiency, so that it can be compared with field measurements in the future. This may be possible, given the paper mentions gaseous pollutants were measured (Line 140). Even without this, it would be helpful to have more of a description of the cookstoves and WBTs which would help with the interpretation of the results.

*Reply:*

As mentioned in the manuscript (page 5, lines 116-118; page 7, lines 151-152), the OC and EC emissions, as well as the absorption of methanol extractable OC from cookstove combustions were reported in our previous work (Xie et al., 2018). In that study, the measurement data of modified combustion efficiency (MCE), overall thermal efficiency (OTE) and emission factors (EFs) of OC and EC for each fuel-cookstove combination were provided in supplementary information.

To make the results of this study comparable to field measurements in the future, we added MCE data for each fuel-cookstove combination in Table S1.

"*Tables S1 and S2 summarized the measurement results of Q$_f$ and Q$_b$, respectively, for each fuel-cookstove combination, including concentrations of carbon contents and light-absorbing properties of sample extracts. As the light absorption of BB BrC is expected to depend largely on burn conditions (Saleh et al., 2014; Pokhrel et al., 2016), the MCE and EC/OC ratio, two indicators of burn conditions, are also given in Table S1.*" (lines 160-165)

The descriptions of the cookstoves and WBT protocol were added when replying to comments 1a and 1b.

**1a.** The stoves are listed in tables in the supplement; however, they are not really discussed in the experimental section of the main paper. How are they different? Where are they used around the world?

*Reply:*

In our previous study (Xie et al., 2018), the light absorption of organic carbon emitted from burning red oak wood and charcoal in cookstoves were investigated using the same samples as this work. That study also provided modified combustion efficiency (MCE) data, overall thermal efficiency (OTE), and emission factors (EFs) of OC and EC for each fuel-cookstove combination during high power phases of the water boiling test, showing the difference across fuel-specific cookstoves.

In the revised manuscript, we added a brief description of each fuel-specific cookstove in supplementary information (Text S1).

"*A brief description of each fuel-specific cookstove was given in supplementary information (Text S1)*" (Lines 141–142)

**1b.** Please include more information about the water boiling tests in the experimental section, as most readers of the journal will not be familiar with it. You should also mention the simmer phase is included for the hot start sample in some tests, if this is correct.

*Reply:*

In the revised manuscript, we added more information on the three test phases in the experimental section.

"*Both CS and HS phases are defined by the duration between the ignition and the water boils. The CS phase starts with the cookstove, pot, and water at ambient temperature; the HS immediately follows the CS with the cookstove hot but the pot and water at ambient temperature; and the SIM phase is defined by a 30-min time period with the cookstove hot and water temperature maintained at 3 ºC below the boiling point.*" (lines 134-139)

The emission test of each fuel-cookstove combination contained a simmer (SIM) phase. Except the 3-stone fire, emission factors (EFs) of OC and EC at the SIM phase were substantially lower than those at high power phases (CS and HS) (Xie et al., 2018). Then, BrC absorption and its molecular composition were primarily measured for CS- and HS-phase samples. In the current work, the SIM-phase samples were analyzed only for red oak burning in a 3-stone fire. This test had comparable OC emissions between CS- and SIM-phase combustions (Xie et al., 2018), and the CS and HS phases of the 3-stone fire are typically similar and cannot be separated. The three SIM-phase samples from the 3-stone fire were treated as HS-phase samples of other cookstove tests. These information on sample selection were originally provided in supplementary information of the manuscript. To make it clear, we moved the information on sample selection to the experiment section of the main text.

"*Details for determinations of OCEC concentrations and BrC absorption were provided in supplementary information (Text S2). Except the 3-stone fire, EFs of OC and EC at the SIM phase were substantially lower than those at high power phases (CS and HS), so the BrC absorption from red oak and charcoal burning were primarily measured for CS- and HS-phase samples in Xie et al. (2018). The SIM-phase samples were analyzed only for red oak burning in a 3-stone fire. This test had comparable OC emissions between CS- and SIM-phase combustions, and CS and HS phases of the 3-stone fire were typically similar and could not be separated (Xie et al., 2018). In the current work, the same emission samples were selected for the analysis of NACs, and the three SIM-phase samples from the 3-stone fire were treated as HS-phase samples of other cookstove tests.*" (Lines 151-160)

**1c.** Could use more reasoning as to how red oak and charcoal are different as seen in Figure S1 C and F by relating hot start and cold start phases to the observed types of combustion. For example, hot start is mostly smoldering for charcoal (high OC emissions with very low BC and therefore low NOx to make less NACs).

*Reply:*

Thanks for the reviewer's suggestions.

In Table S1, the MCE values of charcoal burning indicate that the HS-phase burning is more smoldering than the CS-phase burning. However, the mass ratio of total NACs to OC in percentages (tNAC$_{OC}$%) showed no significant difference ($p$ = 0.29) between HS and CS phases. Considering that the EC/OC ratio of charcoal burning was more sensitive to the initial temperature in the cookstove than MCE variations, it could not be used to predict burn conditions, BrC absorption, or NACs formation from charcoal burning.

Figure S1c and f are used to illustrate the dependence of NACs formation on burn conditions for red oak and charcoal combustions, respectively. Unlike biomass burning, the EC/OC ratio might not be used to parameterize burn conditions of charcoal in cookstoves. We provided a preliminary explanation on the difference of tNAC$_{OC}$% between red oak and charcoal combustions in lines 221-224.

"*Wood burning generates more volatile aromatic compounds (e.g., phenols, PAHs) than charcoal burning (Kim Oanh, et al., 1999), and NACs can form when aromatic compounds and reactive nitrogen (e.g., $NO_X$) are present during solid fuel combustion (Lin et al., 2016, 2017).*"

In comparison to red oak burning, charcoal combustion was more smoldering with significant smaller MCE values ($p$ < 0.01). The wood fire tends to have reduced emissions of $NO_X$ from the smoldering phase (Bertschi et al., 2003). But charcoal and wood are different fuels, and the emission factors (EFs) of $NO_X$ were not measured for controlled cookstove tests in this work. Bhattacharya et al. (2002) reported the EFs of $NO_X$ from a number of traditional and improved cookstoves. They found that EFs for $NO_X$ using wood was slightly lower than charcoal. Then we might not infer that the charcoal burning should emit less $NO_X$ to form NACs in this work.

In the revised manuscript, the original expression

"*Like $MAC_{365}$ and $Å_{abs}$ in $Q_f$ samples for charcoal burning (Xie et al., 2018), tNAC$_{OC}$% derived from the same samples did not correlate with EC/OC ratios in this work (Fig. S1f). Xie et al. (2018) found that the HS-phase for charcoal burning had average OC EFs 5–10 times higher than the CS-phase, while the EC EFs decreased by more than 90% from the CS- to HS-phase, so the EC/OC for charcoal burning is sensitive to the initial temperature in the cookstove, and cannot be used to predict burn conditions, BrC absorption, or NACs formation.*" (lines 253-258)

has been changed into

"*In Table S1, the MCE values of charcoal burning indicate that the HS-phase is more smoldering than the CS-phase. However, the average tNAC$_{OC}$% values showed no significant difference ($p$ = 0.29) between HS and CS phases. Like $MAC_{365}$ and $Å_{abs}$ in $Q_f$ samples for charcoal burning (Xie et al., 2018), tNAC$_{OC}$% derived from the same samples did not correlate with EC/OC ratios in this work (Fig. S1f). Xie et al. (2018) found that the HS-phase for charcoal burning had average OC EFs 5–10 times higher than the CS-phase, while the EC EFs decreased by more than 90% from the CS- to HS-phase. Furthermore, no correlation has been observed between MCE and EC/OC for charcoal burning at the HS-phase. So, the EC/OC for charcoal burning tends to depend more on the initial temperature in the cookstove than MCE variations, and cannot be used to predict burn conditions, BrC absorption, or NACs formation.*" (lines 276-285)

**2.** Regarding source apportionment for NAC measurements (Lines 385-402), these fractions of NAC/OM will be very different in the field because OM can come from many sources. The NAC should be ratioed to a combustion product such as CO or EC.

*Reply:*

In the original manuscript, Figure 2 presents mass fraction patterns of individual NACs in OM from cookstove combustions using red oak wood and charcoal, open biomass burning, and photochemical reactions of typical aromatic precursors with $NO_X$. Receptor models are commonly used for source apportionment of particulate pollutants in the atmosphere (Jaeckels et al., 2007; Shrivastava et al., 2007; Xie et al., 2013), and assume that the ambient pollutants measured in the field are linear combinations from a number of time-variant sources/factors. When using field measurement data of NACs for receptor modeling, the resulting factors can be linked with specific emission sources by comparing with the NAC patterns shown in Figure 2 of this work. Further studies are warranted to unveil NACs patterns of other potential sources (e.g., motor vehicle emissions).

As we mentioned in the introduction, besides combustion sources, atmospheric NACs can also be generated through secondary pathways (lines 106-107). EC is specifically related to primary combustion sources, and CO is totally in the gas phase. In the current work, the gas-phase concentrations of NACs were not available.

To clarify the application of NAC patterns in source apportionment, we added some statement in lines 419-420 and 434-437.

*"This difference among NACs may help with source apportionment using receptor models, which are commonly used and assume that the ambient pollutants measured in the field are linear combinations from a number of time-variant sources/factors. (Jaeckels et al., 2007; Shrivastava et al., 2007; Xie et al., 2013)."*

*"When using field measurement data of NACs for receptor modeling, the resulting factors can be linked with specific emission sources by comparing with the NAC patterns shown in Fig. 2. Further studies are also warranted to unveil NAC patterns of other potential sources (e.g., motor vehicle emissions)."*

**3.** It is implied in lines 412-420 that NACs identified in this study are not significant BrC chromophores, however, if the quartz filter (Qb) is included the fraction is likely higher. It may be more appropriate to give an upper limit given that NACs on Qb could partition into the particle phase in the atmosphere. It is difficult to conclude that NACs are not significant BrC chromophores given the measurements on the sampling artifact that other studies have not considered. Also, NACs may be higher for fuel/stove/cooking activity combinations that result in more flaming combustion which produces NOx, an important reactant for NAC formation. Another factor is that the fractional absorption by NACs was not directly measured. Surrogates were used to quantify NAC concentrations and approximate MACs were used to calculate the Abs365.

*Reply:*

In the manuscript, we mentioned that most identified NACs are strong BrC chromophores, as the average contributions of total NACs to $Abs_{365}$ of sample extracts were more than one order of magnitude higher than their average mass contributions.

*"The average contributions of total NACs to $Abs_{365}$ ($Abs_{365,tNAC}\%$) of the sample extracts ($Q_f$ 1.10 – 2.57%, $Q_b$ 10.7 – 21.0%) are up to 10 times greater than their average $tNAC_{OC}\%$ ($Q_f$ 0.31 – 1.01%, $Q_b$ 1.08 – 3.31%, Table 1). Considering that some NACs are not light-absorbing (Table S4) and the OM/OC ratio is typically greater than*

*unity, most NACs that contribute to Abs$_{365}$ are strong BrC chromophores.*" (Lines 446-450)

Due to the lack of authentic standards, the quantification of NACs concentrations and their contributions to Abs$_{365}$ of Q$_f$ extracts are subject to uncertainties. However, there are evidences showing that BrC absorption is majorly contributed by large molecules with MW > 500 – 1000 Da (Di Lorenzo and Young, 2016; Di Lorenzo et al., 2017). Large NACs molecules may be generated from cookstoves with flaming combustions, and their structures and light absorption are worth future investigations. In previous studies on ambient and biomass burning particles, most identified NACs had a MW lower than 300 – 500 Da, and their total contributions to bulk BrC absorption were estimated to be less than 10% (Mohr et al., 2013; Zhang et al., 2013; Teich et al., 2017; Xie et al., 2019). Similar results were also obtained in the current work. Even if the identified NACs on Q$_b$ are totally derived from evaporation of the upstream filter (negative artifact), the adjusted average contributions of total NACs (Q$_f$ + Q$_b$) to Abs$_{365}$ of Q$_f$ extracts are still lower than 5% (1.59 – 4.01%). Therefore, we suggest that further studies are needed to identify large BrC molecules (including high MW NACs) in ambient and source particles.

The original text from lines 412 to 420 has been changed into

"*All identified NACs explained 1.10 – 2.58% (Fig. S3) of Q$_f$ extracts absorption. Even if the NACs on Q$_b$ were totally derived from upstream filter evaporation, the adjusted average contributions of total NACs (Q$_f$ + Q$_b$) to Abs$_{365}$ of Q$_f$ extracts were still lower than 5% (1.59 – 4.01%). Due to the lack of authentic standards, the quantification of NACs concentrations and their contributions to Abs$_{365}$ of Q$_f$ extracts might be subject to uncertainties. However, growing evidences showed that BrC absorption was majorly contributed by large molecules with MW > 500 – 1000 Da (Di Lorenzo and Young, 2016; Di Lorenzo et al., 2017). Large molecules of NACs may be generated from flaming combustions in cookstoves, and their structures and light absorption are worth future investigations. In previous studies on ambient and biomass burning particles, most identified NACs had a MW lower than 300 – 500 Da, and their total contributions to bulk BrC absorption were estimated to be less than 10% (Mohr et al., 2013; Zhang et al., 2013; Teich et al., 2017; Xie et al., 2019). Similar results were also obtained in the current work. Therefore, further studies are needed to identify large BrC molecules (including high MW NACs) in ambient and source particles.*" (Lines 457-471)

**4.** It is assumed that because Abs365,tNAC% at 365 nm is 7-11 times higher on the quartz fiber backup filter, that NACs may be important light absorbers in the gas phase (lines 442, 425-429, 432-434). To claim this in the paper, more discussion and reasoning for should be given.

**4a.** Those on the backing filter are not necessarily in the gas phase in the natural environment. As you explain in the paper, there are both positive and negative artifacts and there is not likely a straightforward way of calculating what would be in the gas phase.

*Reply:*

Thanks for the reviewer's suggestions. As we mentioned in the original manuscript, the NACs on Q$_b$ were contributed by both positive (gaseous adsorption) and negative (filter evaporation) sampling artifacts. However, the relative contributions

of positive and negative artifacts to $Q_b$ measurements are unknown. Furthermore, gas-phase NACs were not collected using an upstream denuder or an adsorbent cartridge downstream of the filter, and future work is needed to understand the gas/particle distribution of NACs in the ambient and source emissions. Due to the lack of gas-phase NACs data, we overstated that gaseous NACs might be an important group of light-absorbing species in the atmosphere.

The conclusions on light absorption of gaseous NACs has been deleted. Section 3.4 has been reorganized as follows.

"*The average $Abs_{365,iNAC}$% values of $Q_f$ and $Q_b$ samples are presented by fuel type and WBT phase in the Fig. 3 stack plots, and experimental data for each fuel-cookstove are provided in Tables S11–S14. The average contributions of total NACs to $Abs_{365}$ ($Abs_{365,tNAC}$%) of the sample extracts ($Q_f$ 1.10 – 2.57%, $Q_b$ 10.7 – 21.0%) are up to 10 times greater than their average $tNAC_{OC}$% ($Q_f$ 0.31 – 1.01%, $Q_b$ 1.08 – 3.31%, Table 1). Considering that some NACs are not light-absorbing (Table S4) and the OM/OC ratio is typically greater than unity, most NACs that contribute to $Abs_{365}$ are strong BrC chromophores. Like the mass composition of NACs (Fig. 1), $C_{10}H_7NO_3$ (CS 0.24%, HS 0.43%) and $C_8H_9NO_5$ (CS 1.22%, HS 0.55%) were the major contributors to $Abs_{365}$ for the $Q_f$ samples collected during red oak and charcoal burning, respectively (Fig.3a). The average $Abs_{365,tNAC}$% of $Q_b$ samples are 7.53 to 11.3 times higher than those of $Q_f$ samples. Unlike the $Q_f$ samples from red oak burning, $C_{10}H_{11}NO_5$ (CS 2.77%, HS 3.09%) has the highest average contribution to $Abs_{365}$ for $Q_b$ samples, followed by $C_{10}H_7NO_3$ (CS 1.96%, HS 1.32%) and $C_8H_9NO_5$ (CS 1.32%, HS 1.44%). While $C_8H_9NO_5$ dominated the contribution (CS 8.78%, HS 5.82%) to $Abs_{365}$ for the $Q_b$ samples from charcoal burning (Fig. 3b). All identified NACs explained 1.10 – 2.58% (Fig. S3) of $Q_f$ extracts absorption. Even if the NACs on $Q_b$ were totally derived from upstream filter evaporation, the adjusted average contributions of total NACs ($Q_f$ + $Q_b$) to $Abs_{365}$ of $Q_f$ extracts were still lower than 5% (1.59 – 4.01%). Due to the lack of authentic standards, the quantification of NACs concentrations and their contributions to $Abs_{365}$ of $Q_f$ extracts might be subject to uncertainties. However, growing evidences showed that BrC absorption was majorly contributed by large molecules with MW > 500 – 1000 Da (Di Lorenzo and Young, 2016; Di Lorenzo et al., 2017). Large molecules of NACs may be generated from flaming combustions in cookstoves, and their structures and light absorption are worth future investigations. In previous studies on ambient and biomass burning particles, most identified NACs had a MW lower than 300 – 500 Da, and their total contributions to bulk BrC absorption were estimated to be less than 10% (Mohr et al., 2013; Zhang et al., 2013; Teich et al., 2017; Xie et al., 2019). Similar results were also obtained in the current work. Therefore, further studies are needed to identify large BrC molecules (including high MW NACs) in ambient and source particles.*" (Lines 444-471)

**4b.** The vapor pressures of these molecules are very low, and the fraction in the gas phase is low. However, for some nitroaromatics such as 2-nitropenol the vapor pressure is higher. Are the concentrations for some molecules higher on the back up filter compared to the front filter and do we expect them to have higher vapor pressures?

*Reply:*

Due to the lack of measurement data of gas-phase NACs, the gas-phase fractions of NACs are unknown. 4-Nitrophenol (not 2-nitrophenol) was identified and quantified using authentic standards in this work. As the vapor pressure of NACs were rarely

measured or estimated in literatures, the Toxicity Estimation Software Tool (T.E.S.T) developed by the United States Environmental Protection Agency (US EPA) was used to predict subcooled vapor pressure of selected NACs standards at 25 $^o$C ($p^{o,*}_L$) in the following Table.

| Standard compounds | Formula | $m/z$, [M-H]$^-$ | Vapor pressure (atm) |
|---|---|---|---|
| 4-Nitrophenol | $C_6H_5NO_3$ | 138.0196 | $1.58 \times 10^{-5}$ |
| 2-Methyl-4-nitrophenol | $C_7H_7NO_3$ | 152.0353 | $4.57 \times 10^{-6}$ |
| 4-Nitrocatechol | $C_6H_5NO_4$ | 154.0145 | $3.37 \times 10^{-7}$ |
| 2-Methyl-5-nitrobenzoic acid | $C_8H_7NO_4$ | 180.0302 | $1.07 \times 10^{-8}$ |
| 2-Nitro-1-naphthol | $C_{10}H_7NO_3$ | 188.0353 | $4.62 \times 10^{-8}$ |

In comparison to the vapor pressure of $n$-alkanes and polycyclic aromatic hydrocarbons (PAHs) predicted by Xie et al. (2103, 2014), NACs listed in the above table are mostly more volatile than henicosane and fluoranthene ($\sim$10$^{-8}$ atm). Xie et al. (2014) found that the gas-phase concentrations of $n$-alkanes and PAHs with vapor pressure greater than henicosane and fluoranthene were comparable or higher than their particle-phase concentrations. Furthermore, the average $Q_b$ to $Q_f$ mass ratios of the 17 individual NACs ranged from 54.3 ± 24.5% to 135 ± 52.4%, comparable to $n$-alkanes with carbon number ≤ 21 (e.g., henicosane; 26.3 – 163%) and PAHs with benzene ring number ≤ 4 (e.g., fluoranthene; 46.3 – 134%) in the ambient (Xie et al., 2014). So, we suspect that the identified NACs in this study may have substantial fractions remaining in the gas-phase.

In the revised manuscript, more discussions on NACs volatility were added in the last paragraph of section 3.2.

"*In this work, the average $Q_b$ to $Q_f$ mass ratios of the 17 individual NACs ranged from 50.8 ±13.4% to 140 ± 52.9%, comparable to n-alkanes with carbon number ≤ 21 (e.g., henicosane; 26.3 – 163%) and PAHs with benzene ring number ≤ 4 (e.g., fluoranthene; 46.3 – 134%) in the ambient of urban Denver (Xie et al., 2014). Xie et al. (2014) found that the gas-phase concentrations of n-alkanes and PAHs with vapor pressure greater than henicosane and fluoranthene were comparable or higher than their particle-phase concentrations. The vapor pressure of five NACs standards at 25 $^oC$ ($p^{o,*}_L$) were predicted using the US EPA Toxicity Estimation Software Tool (T.E.S.T) and listed in Table S10. Their $p^{o,*}_L$ values are mostly higher than henicosane and fluoranthene ($\sim$10$^{-8}$ atm; Xie et al., 2013, 2014). Then the identified NACs in this study may have substantial fractions remaining in the gas phase.*" Lines (332-341)

**4c.** What are the absorption cross sections for these molecules in the gas phase and their expected gaseous concentrations that would lead us to believe they are significant? Are they long-lived enough in the gas phase to be important? Only solution phase MACs at 365 nm are used to claim that gas phase absorption is significant and this is not sufficient.

*Reply:*

Thanks for the reviewer's comments.

In the current work, NACs from cookstove emissions were identified and quantified using filter samples only. The gas-phase concentrations, absorption cross sections, and life times of identified NACs were not measured or predicted. So, we

overstated that gaseous NACs might be an important group of light-absorbing species in the atmosphere. The conclusions on light absorption of gaseous NACs has been deleted.

**5.** Line 132: Omit that you did kerosene tests. It is not critical as you do not discuss these results.

*Reply:*
The expression in the method section has been changed as suggested. We omitted kerosene tests.

[revised manuscript text omitted]

---

## Author Comment (AC2) · 19 Sep 2020

**Reviewer 2**

This manuscript presents the analysis of N-containing aromatic compounds (NAC) in PM2.5 samples collected from biomass-burning emissions of wood and charcoal in special household stoves. Prior to the HPLC analysis, the collected filters were spiked with one deuterated internal standard and extracted in methanol. The goal of this research was to estimate the contribution of BrC NAC species to the total absorption of PM2.5 samples. The authors also discussed the differences (in OC, total NAC, individual NACs, etc.) between the hot-start and cold-start phases, and also between front and back filters. The authors acknowledged the limitations of this study (e.g., no gas-phase NACs were measured). This study is scientifically important, since NACs are not only light-absorbing compounds, but also are toxic organic species and they are still not well characterized. The manuscript is well organized and well written. I have a few major comments. In summary, I recommend this manuscript for publication after major revisions

*Reply:*
    Thanks for the reviewer's comments, and we'll reply these point by point in the reviewer's specific comments.

**Major comments:**
**1.** The filter samples were spiked with only one deuterated I.S. compound (4-nitrophenol-d4, C6), while the analyzed NACs (Fig 1) have different volatility levels (C7-C11). The author should check if there were potential losses of I.S., which is more volatile than the rest of the analyzed species, and if these losses led to a large overestimates of the concentrations of the analyzed compounds.

*Reply:*
    In this study, the NACs in filter samples were determined identically as Xie et al. (2017, 2019). To minimize the evaporation loss, the sample extract volume was reduced using rotary evaporator under a vacuum, but not nitrogen blowdown evaporation. The coefficient of variation (CV) of the peak area for internal standard (IS) was only 0.16, indicating a stable IS signal. In addition, method recoveries were determined by spiking blank filters with known amounts of standard compounds, followed by extraction and quantification in the same way as that for collected samples. At the end of section 2.2, we mentioned that the average recoveries of NAC standards on pre-baked blank filters ranged from 75.1% to 116% (Lines 197-198). Therefore, the measurement results were not subject to uncertainties due to the loss of internal standard.

**2.** The manuscript contains a lot of abbreviations, which made it very hard to read (HS, CS, Qf, Qb, WBT, OMMs, SIM, etc.)

*Reply:*
    We have defined each abbreviation in the abstract and the rest of the text at the first instance, which satisfied the requirement of the journal.

**3.** Table 1. The concentrations of the total NACs are strikingly high for the backup filters. I am wondering if some sort of unexpected breakthrough happened during the

sampling (especially in the case of charcoal burning). Would it be possible that the BB emissions were quite hot during the sampling, which caused the evaporation from the front filter?

*Reply:*

The mass concentrations of OC and EC were measured for the same filter samples in our previous work (Xie et al., 2018). As no EC has been detected on backup filters ($Q_b$), the breakthrough of particles was not expected during the sampling.

As shown in Table S2, filter samples were mostly collected at ambient temperature (~25 °C). We suspect that the identified NACs in this work have substantial fractions remaining in the gas phase (Lines 332-341).

In lines 344-347, we mentioned that the filter samples were mostly collected near ambient temperature.

"*Considering that most of the $Q_f$ and $Q_b$ samples were collected near ambient temperature (Table S2, ~25 °C), the composition of NACs derived from $Q_f$ measurements alone can be biased due to the lack of gas-phase measurements.*"

4. Some minor comments Line 130. U.S. EPA – please make sure abbreviations are explained in the text Line 131: "USA" should be added after "NC" Line 126, what is "Jiko Poa"? Should company name be added? Lines 152, 156 etc. Company name (+city, state, country) of material and instruments is missing.

*Reply:*

In the revised manuscript, we defined US EPA before it first appeared (Lines 128-129).

"USA" was added after "NC" in line 130.

We added a brief description for all cookstoves, including "Jiko Poa", in supplementary information (Text S1). The company and country names (BURN Manufacturing, Kenya) were added right after "Jiko Poa" (Line 141).

Company, state, and country were added for materials and instruments in lines 169-170, 173.

**References**

Xie, M., Chen, X., Hays, M. D., Lewandowski, M., Offenberg, J., Kleindienst, T. E., and Holder, A. L.: Light Absorption of Secondary Organic Aerosol: Composition and Contribution of Nitroaromatic Compounds, Environmental Science & Technology, 51, 11607-11616, 10.1021/acs.est.7b03263, 2017.

Xie, M., Shen, G., Holder, A. L., Hays, M. D., and Jetter, J. J.: Light absorption of organic carbon emitted from burning wood, charcoal, and kerosene in household cookstoves, Environmental Pollution, 240, 60-67, https://doi.org/10.1016/j.envpol.2018.04.085, 2018.

Xie, M., Chen, X., Hays, M. D., and Holder, A. L.: Composition and light absorption of N-containing aromatic compounds in organic aerosols from laboratory biomass burning, Atmospheric Chemistry and Physics, 19, 2899-2915, 10.5194/acp-19-2899-2019, 2019.

---

## Author Comment (AC3) · 19 Sep 2020

**Reviewer 3**

This manuscript is a nice piece of work describing the emission of nitrogen containing aromatic compounds (NAC) from the use of cookstoves. The authors aim to understand the contribution of these species to the light absorption of organic matter in PM2.5 at the wavelength of 365 nm. The authors found that much higher contribution of NAC light absorption to PM2.5 in quartz fiber backup filters than in PTFE front filters, suggesting that NAC may be an important group of light absorbing compounds in the gas phase. In addition, the authors found the NAC compounds targeted in this manuscript (Mw < 300) are less important for the light absorption of PM2.5 bound organic matter at least from cookstove emissions, and indicate that larger molecules with Mw > 500 are responsible for the light absorption of organic matters. The manuscript is well written, and I recommend publication of this manuscript after addressing minor technical corrections outlined below.

*Reply:*
    Thanks for the reviewer's comments, and we'll reply these point by point in the reviewer's specific comments.

**Specific comments:**
**1.** If it is possible, the authors should present all NAC in emission factors (g kg$^{-1}$) rather than in mass concentrations ($\mu$g m$^{-3}$). Emission factors are more useful than mass concentrations as they can be used for emission control strategies directly.

*Reply:*
    In this work, the target is to characterize the composition, structures, and light absorption of NACs from cookstove emissions. The results will improve our understanding on BrC chromophores and sources. To exhibit NACs composition and absorption, their mass concentrations in filter samples should be provided.
    As we mentioned in the manuscript, the emissions of OC and EC for the same cookstove tests were reported in our previous work (Xie et al., 2018) (lines 150-151). The emission factors (EFs) of total NACs (g kg$^{-1}$ dry fuel) can be obtained from the EFs of OC and the mass ratios of total NACs to OC. In the revised manuscript, the EFs of total NACs were given in Table S5.

**Table S5. Average emission factors of total NACs and OC**

| Fuel &Test phase | Red Oak | | Charcoal | |
|---|---|---|---|---|
| | **CS** | **HS**[a] | **CS** | **HS** |
| *Front filter ($Q_f$)* | | | | |
| Sample number | 18 | 17[b] | 15 | 15 |
| total NAC (mg kg$^{-1}$ dry fuel) | 1.18 ± 0.58 | 1.23 ± 0.69 | 0.79 ± 0.65 | 1.40 ± 0.65 |
| OC (mg kg$^{-1}$ dry fuel) | 244 ± 170 | 340 ± 326 | 179 ± 114 | 619 ± 368 |
| *Backup filter ($Q_b$)* | | | | |
| Sample number | 18 | 17[b] | 14[b] | 15 |
| total NAC (mg kg$^{-1}$ dry fuel) | 0.55 ± 0.24 | 0.61 ± 0.38 | 0.62 ± 0.53 | 1.83 ± 0.79 |
| OC (mg kg$^{-1}$ dry fuel) | 30.5 ± 17.6 | 38.2 ± 24.8 | 66.5 ± 38.9 | 196 ± 96.2 |

[a] Including three SIM phase samples from the 3-stone fire; [b] one filter sample was missed for analysis.

*"The EFs of total NACs shown in Table S5 were obtained by multiplying the EFs of OC and tNAC$_{OC}$%."* (Lines 218-219)

**2.** Line 171: ng _L should be ng/µL or ng µL-1.

*Reply:*

    Revised as suggested. (Line 188)

**3.** Line 214 onwards: This assumes that the backup quartz fiber filter can trap all the gas phase compounds and does not have a breakthrough at all. A better way to estimate the artifact of a filter sampling system is to utilize a denuder in front of a PTFE filter for gas sampling and place a quartz fiber filter after the PTFE to correct a negative artifact from blown-off. It would be good if the authors discuss briefly here about the potential usage of the denuder for the artifact correction in addition to the backup quartz fiber filter.

*Reply:*

    Thanks for the reviewer's suggestions.

    The backup quartz filter was typically used to evaluate the adsorption of gaseous organics ("positive artifact") on filter media. This method does not assume that the backup filter can trap all the gas-phase compounds. We mentioned that the gas-phase NACs were not measured in this study (lines 344-347), and concluded *"Further studies are warranted to investigate the gas/particle distribution of NACs in the ambient and source emissions."* (Lines 481-482)

    In the revised manuscript, we added a few descriptions on the use of a denuder to avoid positive artifacts and its potential issues.

    *"A denuder upstream of the filter for gas sampling was used to avoid positive artifact in several studies (Ding et al., 2002; Ahrens et al., 2012). This approach can generate large negative artifacts by altering the gas-particle equilibrium after the denuder, and a denuder efficiency of 100% might not be guaranteed (Kirchstetter et al., 2001; Subramanian et al., 2004)."* (Lines 238-242)

**4.** Line 283: In Table S3, the authors suggest C8H5NO2 as 3-cyanobenzoic acid. This is commercially available from several chemical suppliers, and it can be positively identified if C8H5NO2 correspond to the authors' suggestion. I ask the authors to purchase the standard compound and quantify it instead of using a surrogate compound.

*Reply:*

    Thanks for the reviewer's suggestion.

    After testing the three isomers of cyanobenzoic acid (2-, 3-, and 4-cyanobenzoic acid), the $C_8H_5NO_2$ molecule from cookstove emissions was identified as 4-cyanobenzoic acid. So, the mass concentration and absorption of $C_8H_5NO_2$ were quantified using 4-cyanobenzoic acid.

    We have updated the tables and figures with $C_8H_5NO_2$ throughout the manuscript. As shown in the figure below, 4-cyanobenzoic acid has no light absorption at > 350 nm. Thus, the contribution of $C_8H_5NO_2$ to Abs$_{365}$ of sample extracts is 0.

[Figure]

**5.** Fig S5: The mass spectrometric conditions and ionization methods used to obtain MS2 spectra for some compounds shown in Fig S5 are very different from ones shown in Fig S4, and they are not comparable at all. I ask the authors to remove those (EI and ESI+) from Fig S5 as they cannot be compared to each other.

*Reply:*

The MS-MS spectra of $C_8H_7NO_4$ and $C_9H_9NO_4$ showed the loss of OCN (Fig. S4g, m), suggesting a structure of benzoxazole/benzisoxazole or the presence of cyanate ($-O-C\equiv N$) or isocyanate ($-O=C=N$) groups. To identify the structure of the OCN group, the MS and MS-MS spectra of four standard compounds, including phenyl cyanate ($C_6H_5OCN$), benzoxazole ($C_7H_5NO$), 4-methoxyphenyl isocyanate ($CH_3OC_6H_4NCO$), and 2,4-dimethoxyphenyl isocyanate $[(CH_3O)_2C_6H_3NCO]$ were obtained from Xie et al. (2019) and shown in Fig. S5 (i-n). These compounds do not have a phenol structure and cannot be detected using ESI under negative ion mode. Fig. S5 (i-n) suggest that the loss of an OCN group only happens during the fragmentation of phenyl cyanate. If Fig. S5 (i-n) is removed, we cannot identify the phenyl cyanate feature for $C_8H_7NO_4$ and $C_9H_9NO_4$ (Lines 380-386).

"*The MS-MS spectra of $C_8H_7NO_4$ eluting at 8.14 min (Fig. S3e) and $C_9H_9NO_4$ eluting at 9.22 min (Fig. S3j) indicate the loss of OCN (Fig. S4g, m), suggesting benzoxazole/benzisoxazole structure or the presence of cyanate ($-O-C\equiv N$) or isocyanate ($-O=C=N$) groups. Mass spectra of selected standard compounds (Fig. S5i-n) in our previous work (Xie et al. 2019) show the loss of an OCN group only happens during the fragmentation of phenyl cyanate. Thus, the $C_8H_7NO_4$ and $C_9H_9NO_4$ isomers containing OCN indicate a phenyl cyanate feature.*"

Xie et al. (2019) identified the phenyl cyanate structure for NACs from open biomass burning in a same way. Therefore, we kept these mass spectra in supplementary information.

**References**

Xie, M., Shen, G., Holder, A. L., Hays, M. D., and Jetter, J. J.: Light absorption of organic carbon emitted from burning wood, charcoal, and kerosene in household cookstoves, Environmental Pollution, 240, 60-67, https://doi.org/10.1016/j.envpol.2018.04.085, 2018.

Xie, M., Chen, X., Hays, M. D., and Holder, A. L.: Composition and light absorption of N-containing aromatic compounds in organic aerosols from laboratory biomass burning, Atmospheric Chemistry and Physics, 19, 2899-2915, 10.5194/acp-19-2899-2019, 2019.